# Ancient medicinal plant rosemary contains a highly efficacious and isoform-selective KCNQ potassium channel opener

Rían W. Manville[1], Derk Hogenkamp[1] & Geoffrey W. Abbott [1]✉

Voltage-gated potassium (Kv) channels in the KCNQ subfamily serve essential roles in the nervous system, heart, muscle and epithelia. Different heteromeric KCNQ complexes likely serve distinct functions in the brain but heteromer subtype-specific small molecules for research or therapy are lacking. Rosemary (*Salvia rosmarinus*) is an evergreen plant used medicinally for millennia for neurological and other disorders. Here, we report that rosemary extract is a highly efficacious opener of heteromeric KCNQ3/5 channels, with weak effects on KCNQ2/3. Using functional screening we find that carnosic acid, a phenolic diterpene from rosemary, is a potent, highly efficacious, $PIP_2$ depletion-resistant KCNQ3 opener with lesser effects on KCNQ5 and none on KCNQ1 or KCNQ2. Carnosic acid is also highly selective for KCNQ3/5 over KCNQ2/3 heteromers. Medicinal chemistry, in silico docking, and mutagenesis reveal that carboxylate-guanidinium ionic bonding with an S4-5 linker arginine underlies the KCNQ3 opening proficiency of carnosic acid, the effects of which on KCNQ3/5 suggest unique therapeutic potential and a molecular basis for ancient neurotherapeutic use of rosemary.

[1] Bioelectricity Laboratory, Dept. of Physiology and Biophysics, School of Medicine, University of California, Irvine, CA, USA. ✉email: abbottg@hs.uci.edu

Voltage-gated potassium (Kv) channels provide a conduit for K$^+$ ions to rapidly diffuse across the plasma membrane in a tightly regulated process essential for cellular excitability and timely cell membrane repolarization. Kv channels in the KCNQ (Kv7) subfamily are extraordinarily diverse in the roles they serve, the tissues in which they are expressed, and the physiological processes they facilitate[1]. This versatility is to a large extent explained by the ability of KCNQ pore-forming α subunits to heteromultimerize, both with each other and with regulatory subunits – especially those of the single-pass transmembrane KCNE subunits. Formation of complexes with KCNE subunits is especially important for KCNQ1, which can form complexes with each of the five KCNE isoforms (1–5) with dramatically different characteristics enabling roles in diverse tissues including the heart, thyroid, pancreas, inner ear, GI tract and choroid plexus[2]. For KCNQ2–5, most of the diversity arises from intrasubfamily heteromerization[3–5], although KCNQ4/5 channels form complexes with KCNE4 in the vasculature, for example[6,7].

In the central nervous system, the primary KCNQ subunits are KCNQ2, 3, and 5, with KCNQ4 thought to have a more limited expression profile, in auditory neurons (and hair cells of the inner ear). KCNQ2/3 heteromers are considered the dominant neuronal KCNQ channel type, and the one most important for generating the neuronal M-current (muscarinic receptor-inhibited current) that is essential for control of neuronal excitability. Indeed, KCNQ2/3 channels act as neuronal gatekeepers, located at the axon initial segment to control whether or not action potentials propagate. Reduced activity of either KCNQ2 or KCNQ3, by loss-of-function mutations in humans, knockout in mice, or pharmacological inhibition, leads to neuronal hyperexcitability and disorders including seizures and developmental delay. KCNQ3/5 channels may also occur in the CNS, and KCNQ2/5 and KCNQ2/3/5 complexes were recently detected using protein chemistry techniques[3–5].

KCNQ2 loss-of-function gene variants are tightly associated with neonatal-onset epileptic encephalopathy, but KCNQ3 and KCNQ5 loss-of-function mutations, and gain-of-function mutations in each of the three, are also associated with epilepsy of varying degrees of severity and developmental delay[8–10].

Understanding neuronal KCNQ isoforms roles in neurological physiology and disease is challenging given the combinatorial complexity of the different possible heteromeric KCNQ complexes in the CNS, their differential spatial and temporal expression, the potentially dynamic nature of their expression and the possibility of homomeric as well as heteromeric KCNQ channels being expressed[4]. Specific small molecules with the ability to distinguish between different KCNQ heteromers in the brain, for research and/or therapeutic purposes, are lacking and highly warranted. We are exploring the potential of plants as chemical factories to provide selective ion channel modulators, often guided by traditional usage of botanical folk medicines[11–16]. Here, we report that rosemary (Salvia rosmarinus), used in traditional medicine for millennia, especially for neurological disorders and to improve memory, is an efficacious neuronal KCNQ channel activator with unique heteromer selectivity, for which we explain the molecular mechanistic and chemical basis.

## Results

### Rosemary extract activates homomeric and heteromeric neuronal KCNQ isoforms

Rosemary is an evergreen flowering plant with needle-shaped, aromatic leaves (Fig. 1a) and clusters of delicate white to pale blue flowers (Fig. 1b). Given the long history of traditional medicinal use of rosemary for a range of purported effects, including sedative, analgesic, anti-paralytic and memory enhancement[17], we tested rosemary extract (1%) for

effects on KCNQ channels, which are highly influential in neural function, first as homomers expressed in Xenopus laevis oocytes. The rosemary extract had no effect on endogenous currents recorded from oocytes injected with water instead of KCNQ cRNA (Fig. 1c), neither did rosemary extract alter KCNQ1 current magnitude (Fig. 1d), KCNQ1 voltage dependence (Fig. 1e, f), or the resting membrane potential ($E_M$) of unclamped oocytes expressing KCNQ1 (Fig. 1g).

Rosemary extract weakly hyperpolarized the activation voltage dependence of homomeric KCNQ2 and KCNQ3* (KCNQ3-A315T, a mutant that permits homomeric KCNQ3 to pass robust currents[18]) (Fig. 1d–f). This effect facilitated a moderate (~10 mV) $E_M$ hyperpolarization by rosemary extract of oocytes expressing KCNQ3 (with a statistically nonsignificant trend for oocytes expressing KCNQ2) (Fig. 1g). Fold increases in current induced by rosemary extract were greatest at −40 to −60 mV and higher in KCNQ3* than for KCNQ2 (Fig. 1h). The shift in voltage dependence of activation induced by rosemary extract was similar for KCNQ2 and KCNQ3* (~−10 mV) (Fig. 1i; Supplementary Table 1).

The moderate effects of rosemary extract on the homomers would not appear sufficient to explain therapeutic effects of rosemary, therefore we next tested extracts from different rosemary plant parts on the predominant M-current-generating heteromer, KCNQ2/3. We found that all three induced essentially similar, minor increases in current at hyperpolarized potentials (Fig. 2a, b), hyperpolarization in the voltage dependence of KCNQ2/3 activation (−7 to −10 mV) (Fig. 2c–e) and hyperpolarization of $E_M$ (Fig. 2f). None of the rosemary parts tested consistently affected activation rate (Fig. 2g) but each of the three consistently slowed deactivation (Fig. 2h). As the effects on KCNQ2/3 heteromers were moderate, we tested KCNQ3/5, a heteromer thought to also generate M-current[19,20]. Unexpectedly, rosemary extract (1/100) strongly opened KCNQ3/5 (Fig. 2i), inducing a −28 mV shift in its voltage dependence of activation, and a shallowing of the slope of the voltage dependence (from 7.9 to 12.1 mV) such that 20% of KCNQ3/5 channels were open at −120 mV (Fig. 2j, k; Supplementary Table 2). This resulted in a −17 mV hyperpolarization of $E_M$, shifting it to the value expected for $E_K$ (Fig. 2l).

### Carnosic acid from rosemary strongly activates KCNQ3*

We next screened compounds previously established to compose rosemary extract[21], to discover the molecular basis for its KCNQ-modulating effects, focusing first on KCNQ3. A screen with 7 compounds (Fig. 3a) each at 100 μM (except hesperidin, which was soluble up to 30 μM) revealed that the phenolic diterpene carnosic acid is a highly efficacious KCNQ3* opener, inducing a −62 mV negative shift in its voltage dependence of activation ($V_{0.5act}$) at 100 μM and promoting robust activity even at −120 mV (Fig. 3b–d). Additionally, hesperidin and quinic acid each induced ~−10 mV shifts in KCNQ3* $V_{0.5act}$ (Fig. 3d; Supplementary Table 3). Only carnosic acid and hesperidin hyperpolarized the $E_M$ of cells expressing KCNQ3* (Fig. 3e).

At 5 μM, carnosic acid robustly potentiated KCNQ3* current at −60 mV upon wash-in, saturating at around 150 s, while washout of the effect was much slower (Fig. 4a). The $EC_{50}$ for current increase at −60 mV was $5.43 \pm 0.37$ μM (Fig. 4b); the $EC_{50}$ for $\Delta V_{0.5act}$ was $18 \pm 0.72$ μM (Fig. 4c) and the $EC_{50}$ for shifting the $E_M$ of unclamped oocytes expressing KCNQ3* was $1.19 \pm 0.19$ μM (Fig. 4d). Carnosic acid voltage-independently doubled the activation rate (Fig. 4e, f) and voltage-dependently slowed the deactivation rate ( >fourfold at −120 mV) (Fig. 4g, h) of KCNQ3*. In contrast, carnosic acid was only weakly active against KCNQ2 and KCNQ2/3 across the concentration range

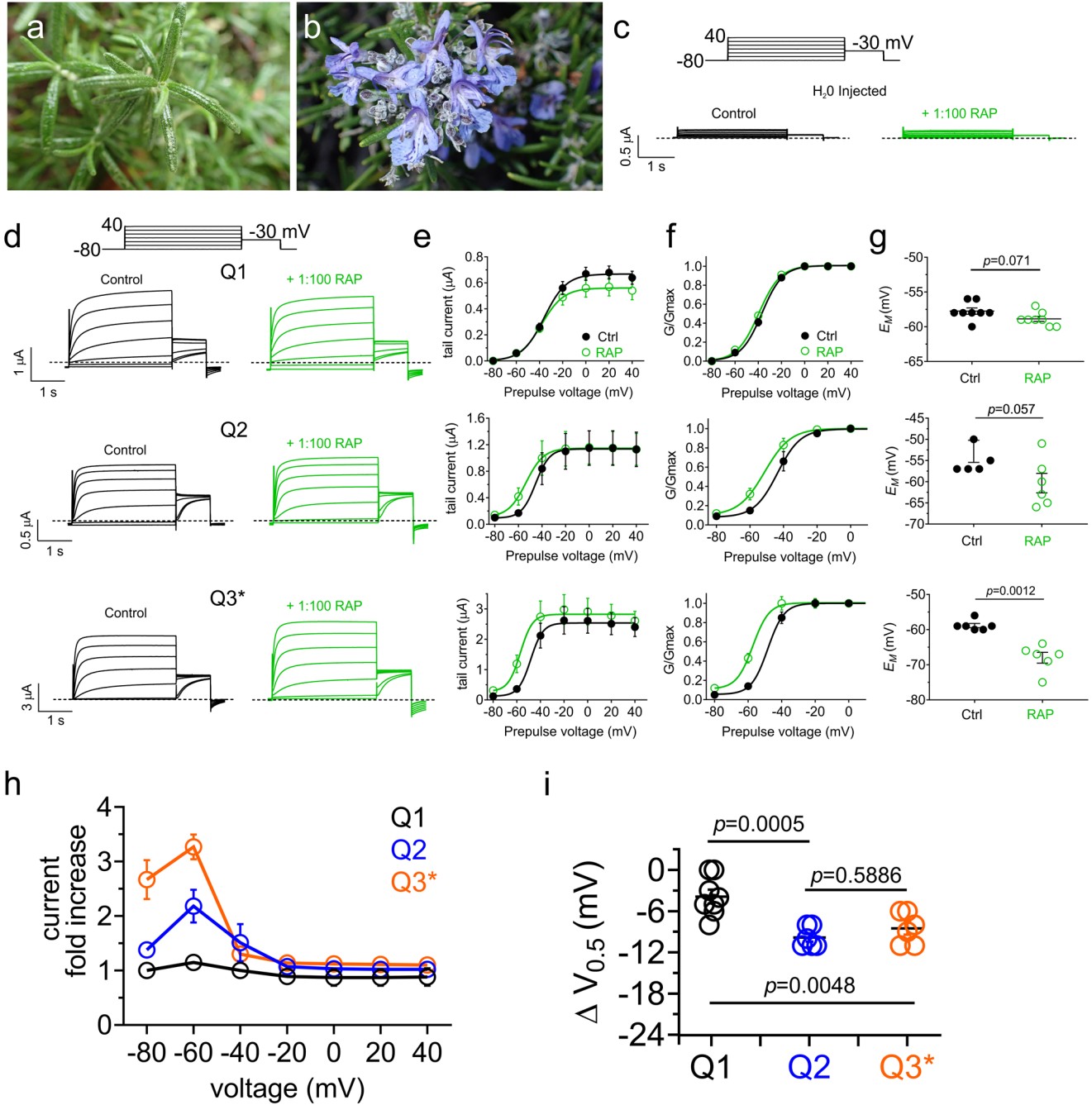

**Fig. 1 Rosemary extract negative-shifts the voltage dependence of homomeric KCNQ2 and KCNQ3 channels. a** *Salvia rosmarinus* leaves (image: Bo Abbott, used by permission). **b** *Salvia rosmarinus* flowers (image: GWA). **c** Mean traces for water-injected oocytes in the absence (Control) or presence of 1:100 RAP extract. Dashed lines indicate zero current line here and throughout. Scale bars lower left for each trace; voltage protocol upper inset; $n = 5$ per group. **d** Mean traces for KCNQ1-3 homomers expressed in oocytes in the absence (Control) or presence of 1:100 RAP extract. Scale bars lower left for each trace; voltage protocol upper inset; $n = 5$–10 per group. **e** Mean tail current for traces as in (**d**); $n = 5$–10 per group. **f** Mean normalized tail current (G/Gmax) for traces as in (**d**); $n = 5$–10 per group. **g** Mean unclamped oocyte membrane potential for KCNQ1-3 homomer-expressing oocytes as in d; $n = 5$ per group. **h** Mean current fold-increase versus membrane potential for traces as in d in response to 1:100 RAP extract; $n = 5$–10 per group. **i** Mean $\Delta V_{0.5}$ activation for traces as in (**d**) in response to 1:100 RAP extract; $n = 5$–10 per group. Error bars indicate SEM. $n$ indicates number of biologically independent oocytes. Statistical comparisons by paired $t$-test or one-way ANOVA. RAP rosemary aerial parts.

tested (Fig. 4i–n; Supplementary Table 4). Thus, carnosic acid is highly selective for KCNQ3 over KCNQ2, and the KCNQ2 reduced sensitivity to carnosic acid is dominant in KCNQ2/3 heteromers.

**Carnosic acid activation of KCNQ3* is resilient to PIP$_2$ reduction.** Under baseline conditions, binding of the soluble,

lipid-derived signaling molecule phosphatidylinositol 4,5-bisphosphate (PIP$_2$) to KCNQ3 (and other KCNQ isoforms) is required for efficient coupling of the voltage sensor to pore opening and thus voltage-dependent gating[22,23]. Here, reduction of PIP$_2$ levels using pretreatment with wortmannin reduced KCNQ3* current magnitude ~fivefold, yet carnosic acid (5 μM) was still able to shift the KCNQ3* $V_{0.5act}$ (Fig. 5a, b) and $E_M$

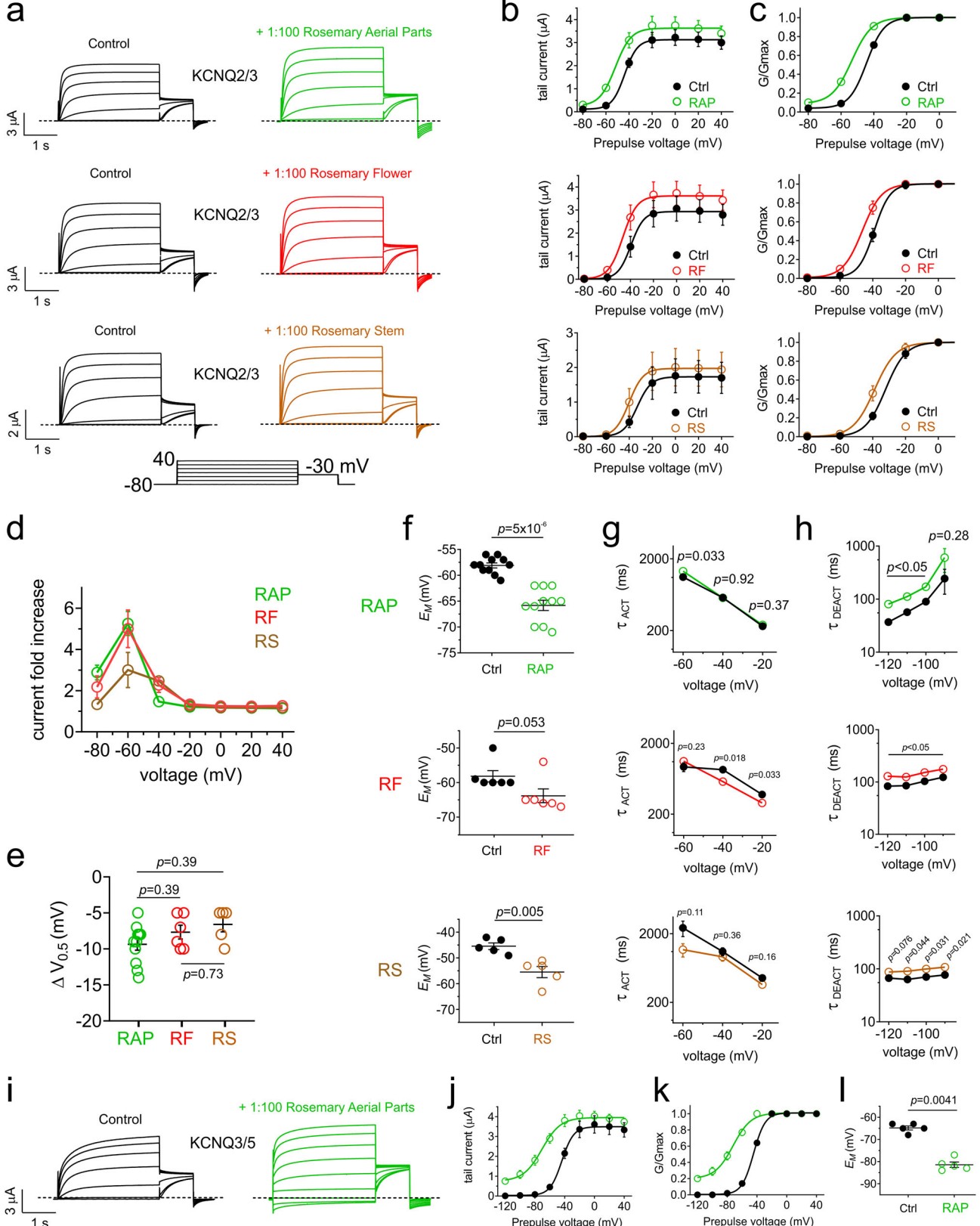

(Fig. 5c) under these conditions, albeit the $\Delta V_{0.5act}$ was ~50% of that under PIP₂ replete conditions (Fig. 5d). Another strategy for PIP₂ reduction is to express a voltage-sensitive phosphatase (VSP) that reduces PIP₂ levels upon membrane depolarization[24]. Here, we coexpressed the *Danio rerio* VSP (DrVSP)[25] with KCNQ3* and found that after 12 s at

+120 mV, the KCNQ3* current decayed by 50%. Strikingly, carnosic acid (5 μM) reduced or prevented DrVSP-induced KCNQ3* current decay under similar conditions (Fig. 5e, f; Supplementary Table 5). Interestingly, we previously observed that GABA binding is able to open KCNQ2/3 channels after wortmannin pretreatment[26].

**Fig. 2 Rosemary extract is a more efficacious opener of heteromeric KCNQ3/5 versus KCNQ2/3 channels. a** Mean traces for KCNQ2/3 heteromers expressed in oocytes in the absence (Control) or presence of 1:100 extract of rosemary parts as indicated. Scale bars lower left for each trace; voltage protocol lower inset; $n = 5–11$ per group. **b** Mean tail current for KCNQ2/3 traces as in a; $n = 5–11$ per group. **c** Mean normalized tail current (G/Gmax) for KCNQ2/3 traces as in a; $n = 5–11$ per group. **d** Mean current fold-increase versus membrane potential for KCNQ2/3 traces as in A in response to 1:100 extracts as indicated; $n = 5–11$ per group. **e.** Mean $\Delta V_{0.5}$ activation for KCNQ2/3 traces as in (**a**) in response to 1:100 extracts as indicated; $n = 5–11$ per group. **f** Mean unclamped oocyte membrane potential for KCNQ2/3-expressing oocytes as in (**a**); $n = 5–11$ per group. **g** Mean activation rate for KCNQ2/3 traces as in (**a**) in response to 1:100 extracts as indicated; $n = 5–11$ per group. **h** Mean deactivation rate for KCNQ2/3 traces as in (**a**) in response to 1:100 extracts as indicated; $n = 5–11$ per group. **i** Mean traces for KCNQ3/5 heteromers expressed in oocytes in the absence (Control) or presence of 1:100 extract of rosemary aerial parts. Scale bars lower left; voltage protocol as in (**a**); $n = 5$. **j** Mean tail current for KCNQ3/5 traces as in (**i**); $n = 5$. **k** Mean normalized tail current (G/Gmax) for KCNQ3/5 traces as in (**i**); $n = 5$. **l** Mean unclamped oocyte membrane potential for KCNQ3/5 heteromer-expressing oocytes as in (**i**); $n = 5$. Error bars indicate SEM. $n$ indicates number of biologically independent oocytes. Statistical comparisons by paired $t$-test or one-way ANOVA. RAP rosemary aerial parts, RF rosemary flower, RS rosemary stem.

Of the other rosemary compounds tested (Fig. 6a), only hesperidin opened KCNQ2/3 channels ($-10.4$ mV $V_{0.5act}$ shift at 30 μM) (Fig. 6b–d) and hyperpolarized the $E_M$ of KCNQ2/3-expressing oocytes (Fig. 6e) yet hesperidin did not open KCNQ2 (Fig. 6f–i; Supplementary Table 6), indicating that in contrast to carnosic acid, hesperidin effects on KCNQ3* (Fig. 3) are dominant in KCNQ2/3 complexes. The weak opening effects of both hesperidin and carnosic acid likely underlie the weak KCNQ2/3-opening ability of rosemary extract.

**Carnosic acid is a potent and efficacious KCNQ3/5 channel opener.** Carnosic acid also activated KCNQ5, inducing a relatively small ($-7.2$ mV) shift in $V_{0.5act}$ (at 100 μM) but, more notably, a component ($\sim 5\%$) of constitutively activated current even at $-120$ mV (Fig. 7a–c). Strikingly, carnosic acid (100 μM) was a powerful opener of KCNQ3/5 heteromeric channels and also overcame the relative insensitivity of KCNQ2 that dominated the response of KCNQ2/3 heteromers (Fig. 4) to negative-shift the voltage dependence of activation of KCNQ2/5 and KCNQ2/3/5 heteromers, both of which, like KCNQ3/5, are thought to contribute to neuronal M-current[5] (Fig. 7a–c). Accordingly, carnosic acid hyperpolarized $E_M$ in oocytes expressing KCNQ5-containing homomers and heteromers (Fig. 7d). Carnosic acid dose response studies revealed that effects on KCNQ3/5 were intermediate between those on KCNQ3 and KCNQ5 homomers (Fig. 7e, f), as were effects on KCNQ2/5 and KCNQ2/3/5 (Fig. 7g, h) (Supplementary Table 7). Thus, carnosic acid likely underlies the KCNQ3/5 opening ability of rosemary extract (Fig. 2i–l).

**Carnosic acid interaction is facilitated by two arginines on the KCNQ3 S4-5 linker.** Because of the robust opening effects of carnosic acid on KCNQ3 and KCNQ3/5 channels, and our previous findings of synergy between small molecules that preferentially target different isoforms in KCNQ heteromers[12], we tested the combination of carnosic acid and the KCNQ5-selective opener, aloperine[15] (Fig. 8a) on KCNQ3/5. Aloperine (5 μM) was a weak KCNQ3/5 opener, while subsequent addition of carnosic acid (5 μM) in the presence of aloperine paradoxically further weakened the opening effect (Fig. 8b, c) and the ability to induce KCNQ3/5 hyperpolarization of $E_M$ (Fig. 8d; Supplementary Table 8). This suggested steric hindrance between the two small molecules, consistent with a similar interaction site for the two. We previously found that aloperine binds close to R212 in the KCNQ5 S4-5 linker[15]. Here, carnosic acid in silico-docked close to the equivalent arginine and its neighbor on the cryo-EM-derived structure of human KCNQ2 (R213, R214)[27] and to the AlphaFold[28,29] model of human KCNQ3 (R242, R243) (Fig. 8e). Interestingly, the in silico docking predicted ionic bond formation between carnosic acid via its carboxylate group to the guanidinium group of R243 in KCNQ3, but not in KCNQ2 (Fig. 8e; closeup in Fig. 8f). We do not yet understand what about the

KCNQ3 versus the KCNQ2 environment around R243 influences this predicted difference in interaction with carnosic acid.

To test the predictions, we first conducted alanine-scanning mutagenesis of KCNQ3-R242, R243 and nearby residues, and found that mutation of R242 or R243 greatly diminished effects of carnosic acid (100 μM), in particular the ability to open KCNQ3* at strongly hyperpolarized potentials (Fig. 8g, h; Supplementary Table 8; for comparison of mutant versus wild-type baseline voltage dependence, see Supplementary Fig. 1) although carnosic acid retained the ability to hyperpolarize $E_M$ because this occurs at less hyperpolarized potentials (Fig. 8i). A dose-response study showed that the R242A mutation halved the efficacy of carnosic acid in terms of the $\Delta V_{0.5act}$, (Fig. 8j) while we were not able to unambiguously interpret the dose response for the $\Delta E_M$ shift as it did not saturate at the highest possible carnosic acid dose (Fig. 8k).

Interestingly, mutation to alanine of R239 or D241 resulted in nonfunctional channels, suggesting the importance of these residues in channel activation or possibly protein folding. KCNQ3*-M240A had a positive-shifted voltage dependence of activation compared with KCNQ3* but was still strongly opened by carnosic acid (100 μM), with 20% constitutive current at $-120$ mV. KCNQ3*-G244A behaved indistinguishably from KCNQ3* with respect to both baseline activity and carnosic acid effects. KCNQ3*-G245A midpoint voltage dependence of activation was shifted $-32$mV compared to KCNQ3*, yet still showed robust sensitivity to carnosic acid ($-45$ mV shift in midpoint voltage dependence of activation with 100 μM carnosic acid (Fig. 8g, h; Supplementary Table 8).

We next tested the effects on carnosic acid opening of mutating KCNQ3-W265, a residue essential for retigabine and GABA binding to KCNQ3[26,30,31], and also P211 and L198, residues identified by others as important for determining differential selectivity of ICA069673 for KCNQ2 vs KCNQ3[32]. Carnosic acid displayed similar opening efficacy for KCNQ3* and KCNQ3*-W265L channels, although the latter was approximately fivefold more sensitive in terms of potency (Fig. 9a–c). KCNQ3*-P211A and KCNQ3*-L198F channels behaved very similarly to KCNQ3* in terms of carnosic acid efficacy and potency (Fig. 9a–c).

Comparing shifts in $V_{0.5act}$, R242A, and R243A most strongly reduced efficacy of carnosic acid, followed by mutation of either of three proximal residues (M240A, G244A, G245A) (Fig. 9d; Supplementary Table 9). Overall, the data strongly suggest that KCNQ3 R242 and R243 are important components of carnosic acid interaction and/or its functional effects.

**Carnosic acid has optimal chemical characteristics for KCNQ3 opening.** The in silico docking and in vitro mutagenesis studies (Figs. 8 and 9) suggest that carnosic acid activation of KCNQ3* is facilitated by ionic bonding between the carboxylate group on carnosic acid and the guanidinium group on KCNQ3*-R243

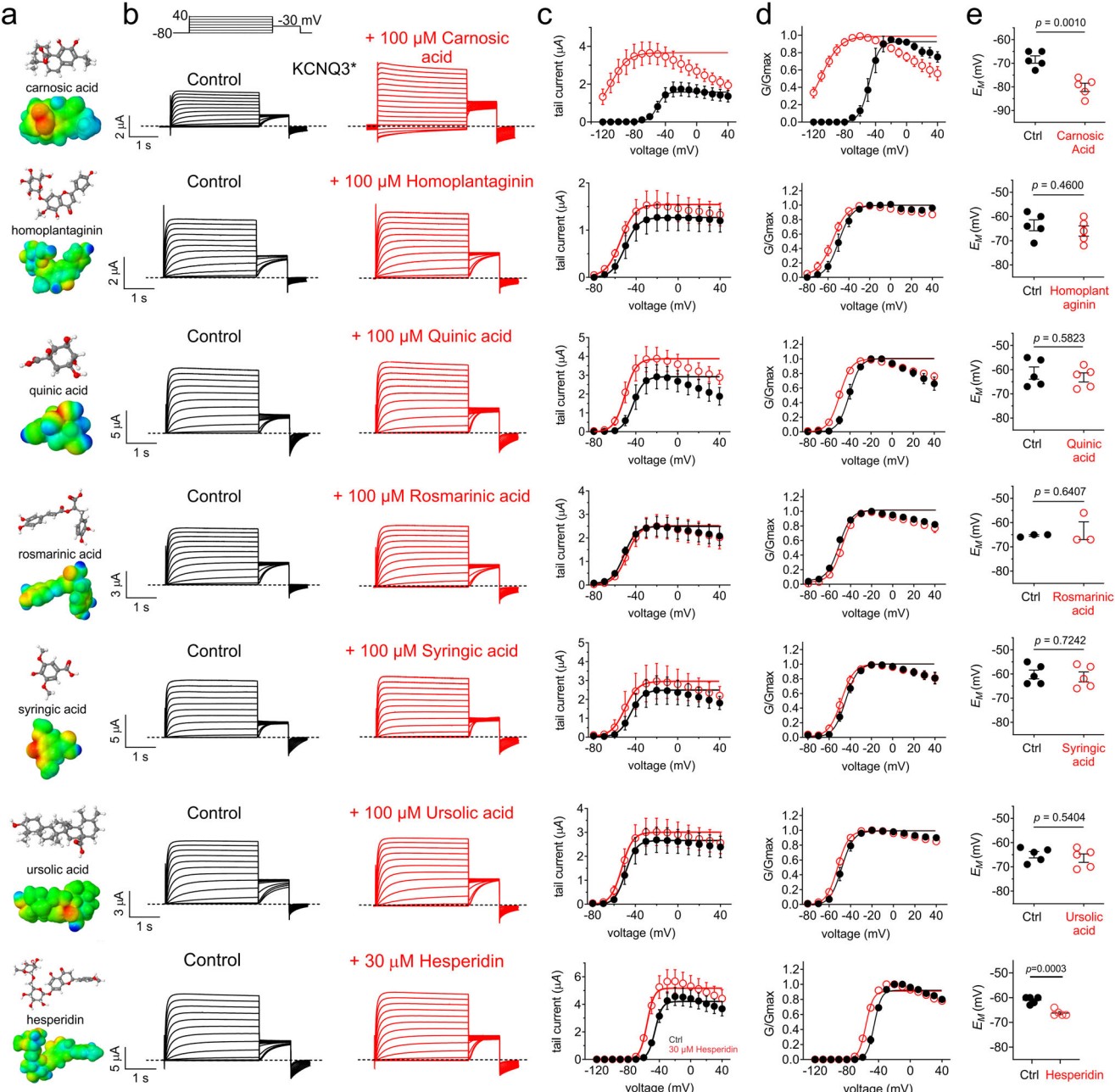

**Fig. 3 Carnosic acid from rosemary is a highly efficacious KCNQ3 opener. a** Jmol plots of structure (upper) and surface charge (lower) for rosemary compounds as indicated. **b** Mean traces for KCNQ3* homomers expressed in oocytes in the absence (Control) or presence of 30–100 µM rosemary compounds as indicated. Scale bars lower left for each trace; voltage protocol upper inset; $n = 3$–5 per group. **c** Mean tail current for KCNQ3* traces as in (**b**); $n = 3$–5 per group. **d** Mean normalized tail current (G/Gmax) for KCNQ3* traces as in (**b**); $n = 3$–5 per group. **e** Mean unclamped oocyte membrane potential for KCNQ3*-expressing oocytes as in (**b**); $n = 3$–5 per group. Error bars indicate SEM. $n$ indicates number of biologically independent oocytes. Statistical comparisons by paired $t$-test or one-way ANOVA.

(Fig. 10a, b). We next in silico docked carnosic acid to mutant KCNQ3 (R242A and/or R243A) and found that the single mutations diminished the predicted binding free energy (ΔG) and that it was further reduced by the double mutation; R243A and R242A,R243A mutations also eliminated the predicted ionic bond formation described above (Fig. 10b). To further test the importance of these interactions, we synthesized and tested several compounds related to carnosic acid for their KCNQ opening activity. Carnosic acid exhibits negative electrostatic surface potential centered on the carboxylate group predicted to facilitate binding to KCNQ3 (Fig. 10c). Carnosol, also present in rosemary[21], lacks the carboxylate group present in carnosic acid

yet exhibits predicted negative electrostatic surface potential on the same side of the molecule as for carnosic acid, but not exactly the same location and this is countered by predicted positive potential on the distal side of the molecule (Fig. 10c). Carnosol was thus not predicted to form an ionic bond to the R243 guanidinium group (Fig. 10d) and at 100 µM produced only a weak, −11-mV negative shift in KCNQ3* $V_{0.5act}$ (compared with −62 mV negative shift for carnosic acid at 100 µM; Fig. 3c, d) and no shift for KCNQ2 or KCNQ2/3 (Fig. 10e, f). Similarly, carnosol did not alter $E_M$ of oocytes expressing KCNQ2, KCNQ3*, or KCNQ2/3 (Fig. 10g). We next added two methyl groups to carnosol, creating dimethyl carnosol, which has a surface charge

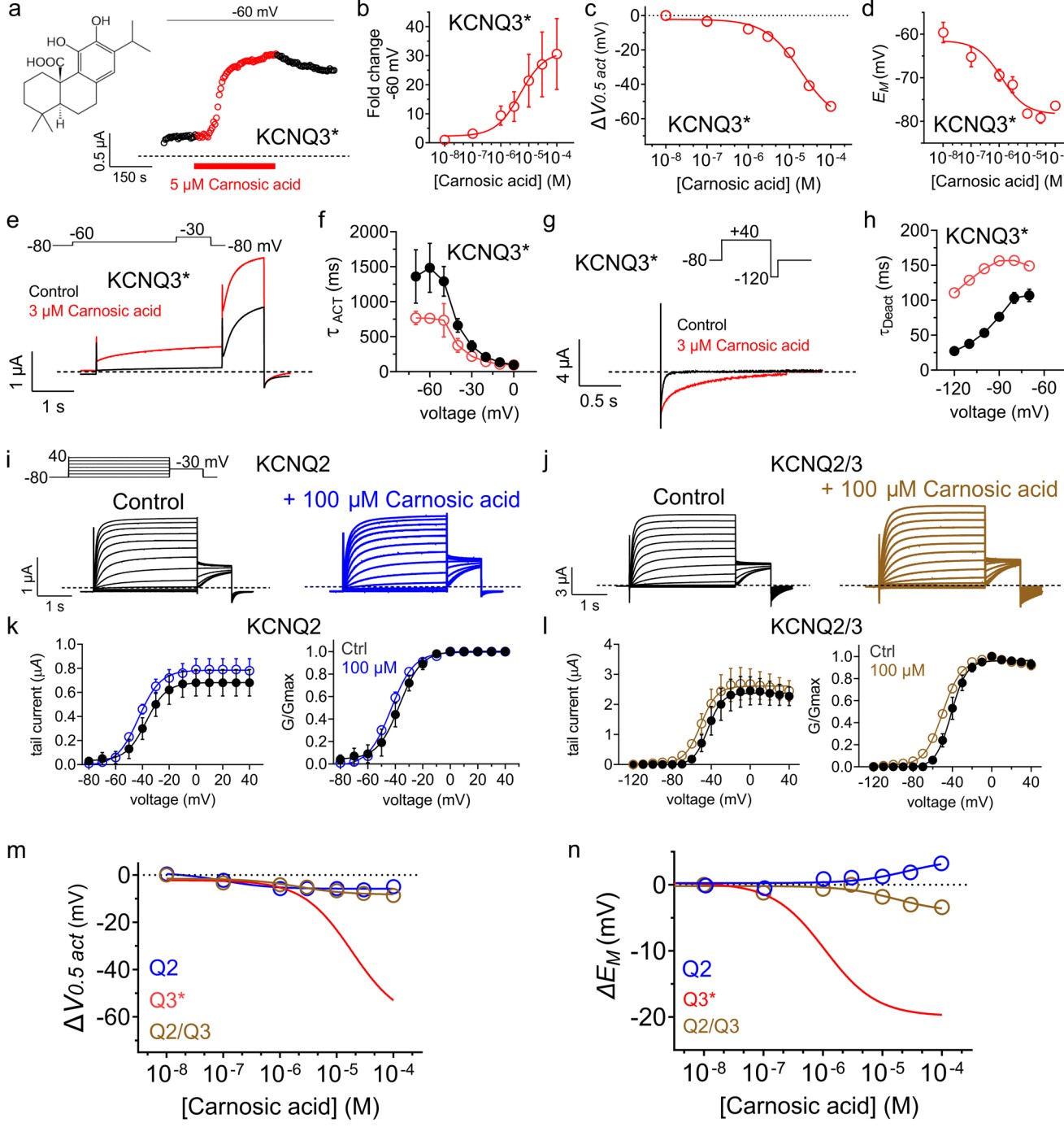

profile somewhat more similar to that of carnosic acid (Fig. 10c), but cannot form an ionic bond to R243 (Fig. 10d), and only induced a −3.2-mV hyperpolarizing shift in KCNQ3* $V_{0.5act}$ (Fig. 10e, f), although it was able to hyperpolarize $E_M$ by >8 mV (Fig. 10g).

We next synthesized carnosic acid γ-lactone, which does not contain the carboxylate group but is predicted to H-bond to R243 via a phenolic group at the other end of the molecule (Fig. 10h, i). Interestingly, carnosic acid γ-lactone was able to hyperpolarize the $V_{0.5act}$ of both KCNQ3* and KCNQ2/3, albeit very weakly (−8.7 and −10 mV, respectively) (Fig. 10j, k) and without hyperpolarizing $E_M$ (Fig. 10l) (note that because of solubility issues we could only test it at 10 μM). Similarly, methyl carnosate, in which we replaced the carnosic acid carboxylate group with a methyl ester group (Fig. 10h), was not predicted to form an ionic

bond to either arginine in KCNQ3 (Fig. 10i), was only a weak activator of KCNQ3* and did not open KCNQ2/3, albeit we could only test at 10 μM because of poor solubility (Fig. 10m–o).

Finally, we focused on carnosic acid derivatives that retained the carnosic acid carboxylate group and its predicted negative electrostatic surface potential location, but because of their altered structures, also had positive electrostatic surface potential at the opposite end of the molecule to the carboxylate group. Thus, pisiferic acid had a predicted free energy of binding to the KCNQ3 S4/5 arginines that was very similar to that of carnosic acid (−8.1 versus −8.2 kcal/mol, respectively), but had a different predicted binding orientation to that of carnosic acid, lacked the predicted ionic bonding (Fig. 10p, q) and only weakly negative-shifted the voltage dependence of KCNQ3* at 100 μM (Fig. 10r–t). Carnosic acid diacetate had a similar predicted

**Fig. 4 Carnosic acid potently opens KCNQ3\* but not KCNQ2 or KCNQ2/3. a** Exemplar plot showing wash-in (red) and partial washout (black) of carnosic acid (5 µM) at −60 mV on KCNQ3\* expressed in oocytes. Upper inset, carnosic acid structure. **b** Mean current fold increase versus [carnosic acid] at −60 mV for KCNQ3\*; $n = 5$ per group. **c** Mean $\Delta V_{0.5act}$ versus [carnosic acid] for oocytes expressing KCNQ3\*; $n = 5$ per group; voltage protocol as in Fig. 3b. **d** Mean $E_M$ versus [carnosic acid] for unclamped oocytes expressing KCNQ3\*; $n = 5$ per group. **e** Mean traces for KCNQ3\* showing effect of carnosic acid (3 µM) (red) on activation. Scale bars lower left; voltage protocol upper inset; $n = 5$ per group. **f** Mean activation rate versus voltage for KCNQ3\* traces as in f across the voltage range in the absence (black circles) or presence (open red circles) of 3 µM carnosic acid; $n = 5$ per group. **g** Mean traces for KCNQ3\* showing effect of carnosic acid (3 µM) on deactivation. Scale bars lower left for each trace; voltage protocol upper inset; $n = 8$ per group. **h** Mean deactivation rate versus voltage for KCNQ3\* traces as in g in the absence (black circles) or presence (open red circles) of 3 µM carnosic acid; $n = 8$ per group. **i** Mean traces for KCNQ2 expressed in oocytes in the absence (Control) or presence (blue) of 100 µM carnosic acid. Scale bars lower left; voltage protocol upper inset; $n = 5$ per group. **j** Mean traces for KCNQ2/3 expressed in oocytes in the absence (Control) or presence (brown) of 100 µM carnosic acid. Scale bars lower left; voltage protocol upper inset; $n = 5$ per group. **k** Mean tail current and normalized tail current (G/Gmax) for KCNQ2 traces as in (**l**); $n = 5$ per group. **l** Mean tail current (left) and normalized tail current (G/Gmax) (right) for KCNQ2/3 traces as in (**j**); $n = 5$ per group. **m** Mean $\Delta V_{0.5act}$ versus [carnosic acid] for oocytes expressing KCNQ2 or KCNQ2/3, with KCNQ3\* data from panel c for comparison; $n = 5$ per group; voltage protocol as in (**l**). **n** .Mean $\Delta E_M$ versus [carnosic acid] for oocytes expressing KCNQ2 or KCNQ2/3, with KCNQ3\* data (calculated from panel d) for comparison; $n = 5$ per group; voltage protocol as in panel i. Error bars indicate SEM. $n$ indicates number of biologically independent oocytes.

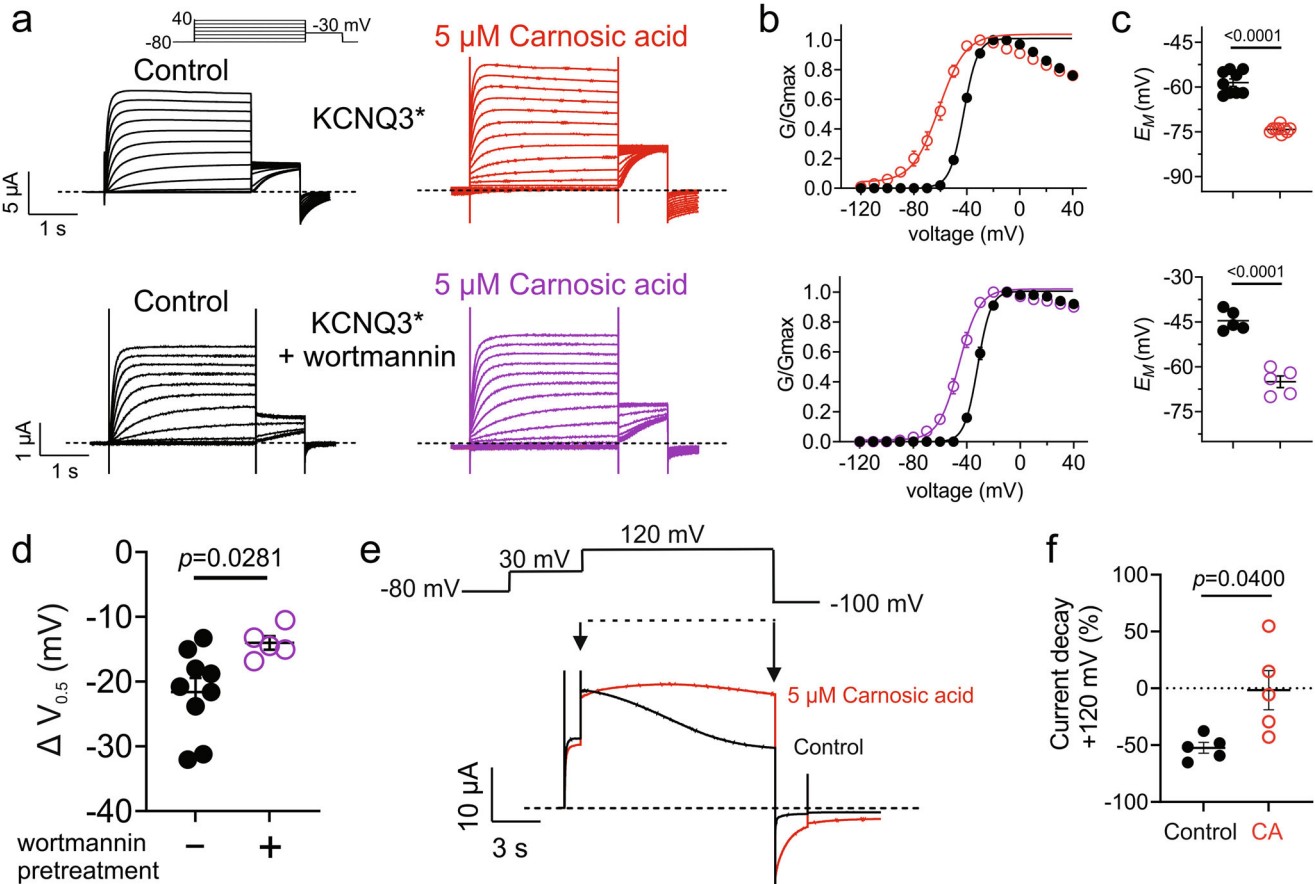

**Fig. 5 Carnosic acid KCNQ3\* opening is resilient to PIP₂ reduction. a** Mean traces for KCNQ3\* expressed in oocytes in the absence (Control) or presence of carnosic acid (5 µM) with or without wortmannin pretreatment to deplete PIP₂. Scale bars lower left for each trace; voltage protocol upper inset; $n = 5$–9 per group. **b** Mean normalized tail current (G/Gmax) for KCNQ3\* traces as in a; $n = 5$–9 per group. **c** Mean unclamped oocyte membrane potential for KCNQ3\*-expressing oocytes as in a; $n = 5$–9 per group. **d** $\Delta V_{0.5act}$ induced by carnosic acid (5 µM) for KCNQ3\* expressed in oocytes as in (**a**), in the absence (black) or presence (purple) of wortmannin pretreatment; $n = 5$–9 per group. **e** Mean traces for KCNQ3\* coexpressed in oocytes with DrVSP in the absence (black; Control) or presence (red) of carnosic acid (5 µM), pulsed to +120 mV to activate VSP and deplete PIP₂; voltage protocol, upper inset; $n = 5$. **f** Mean current decay for period between arrows in panel e for KCNQ3\* in the absence (Control) or presence (CA) of carnosic acid (5 µM); $n = 5$. Error bars indicate SEM. $n$ indicates number of biologically independent oocytes. Statistical comparisons by t-test or one-way ANOVA.

electrostatic surface potential balance to that of pisiferic acid (Fig. 10p), adopted a similar predicted, non-ionic bonded orientation relative to the KCNQ3 arginines to that of pisiferic acid (Fig. 10q) and only weakly negative-shifted the voltage dependence of KCNQ3\* or KCNQ2/3 (note that because of solubility issues we could only test it at 10 µM) (Fig. 10r–t). In summary, carnosic acid is a Goldilocks compound in this chemical space, with an essential carboxylate group and charge balance highly optimal for KCNQ3\* opening (Fig. 10u, v; Supplementary Table 10).

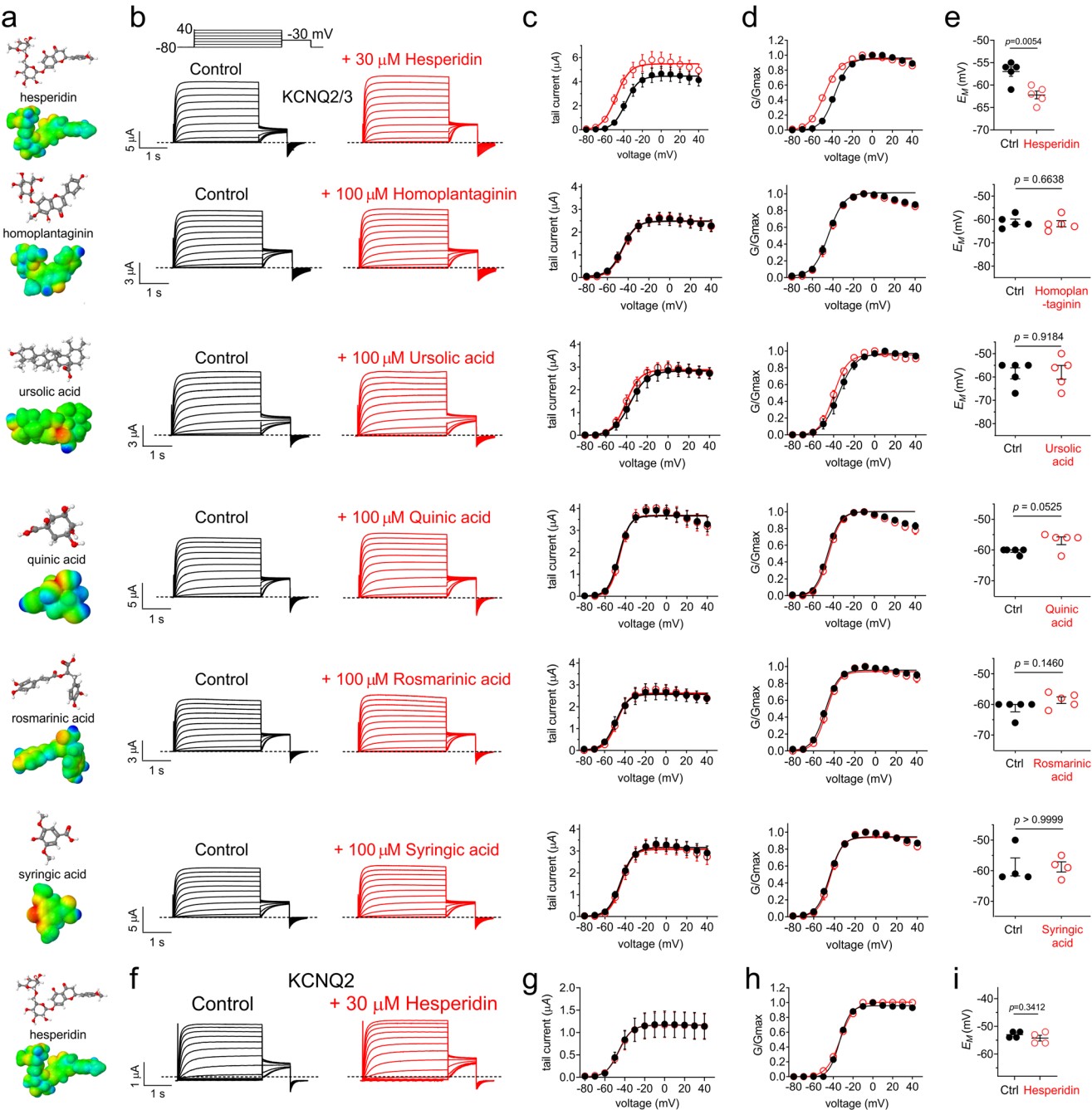

**Fig. 6 Hesperidin weakly activates KCNQ2/3 but not KCNQ2. a** Jmol plots of structure (upper) and surface charge (lower) for rosemary compounds as indicated. **b** Mean traces for KCNQ2/3 heteromers expressed in oocytes in the absence (Control) or presence of 30 or 100 µM rosemary compounds (red) as indicated. Scale bars lower left; voltage protocol upper inset; $n = 4$–5 per group. **c** Mean tail current for KCNQ2/3 traces as in (**b**); $n = 4$–5 per group. **d** Mean normalized tail current (G/Gmax) for KCNQ2/3 traces as in (**b**); $n = 4$-5 per group. **e** Mean unclamped oocyte membrane potential for KCNQ2/3-expressing oocytes as in b; $n = 4$–5 per group. (**f**) Mean traces for KCNQ2 homomers expressed in oocytes in the absence (Control) or presence of 30 µM hesperidin (red). Scale bars lower left; voltage protocol upper inset; $n = 4$ per group. (**g**) Mean tail current for KCNQ2 traces as in (**f**); $n = 4$ per group. (**h**) Mean normalized tail current (G/Gmax) for KCNQ2 traces as in (**f**); $n = 4$ per group. **i** Mean unclamped oocyte membrane potential for KCNQ2-expressing oocytes as in (**f**); $n = 4$ per group. Error bars indicate SEM. $n$ indicates number of biologically independent oocytes. Statistical comparisons by one-way ANOVA.

## Discussion

Rosemary is a rich source of flavonols, phenolic acids, terpenoids and essential oils[21], and has traditionally been used medicinally for a wide variety of disorders, ranging from cancer to gastrointestinal problems to memory loss, as well as widespread use for its flavor in cuisines around the world[17,33]. Rosemary derives its name from the Latin for dew of the sea (*ros marinus*) because it is

native to coastal cliffs of the Mediterranean as well as Asia; its range now reaches as far as the Americas, England and North Africa. Rosemary was revered in Ancient Greece, Rome, Israel and Egypt for its purported beneficial uses against memory loss, muscle ache, infection and gastrointestinal dysfunction. The Romans are thought to have introduced it to the British Isles, where it was used by the Celtic Druids for indications such as

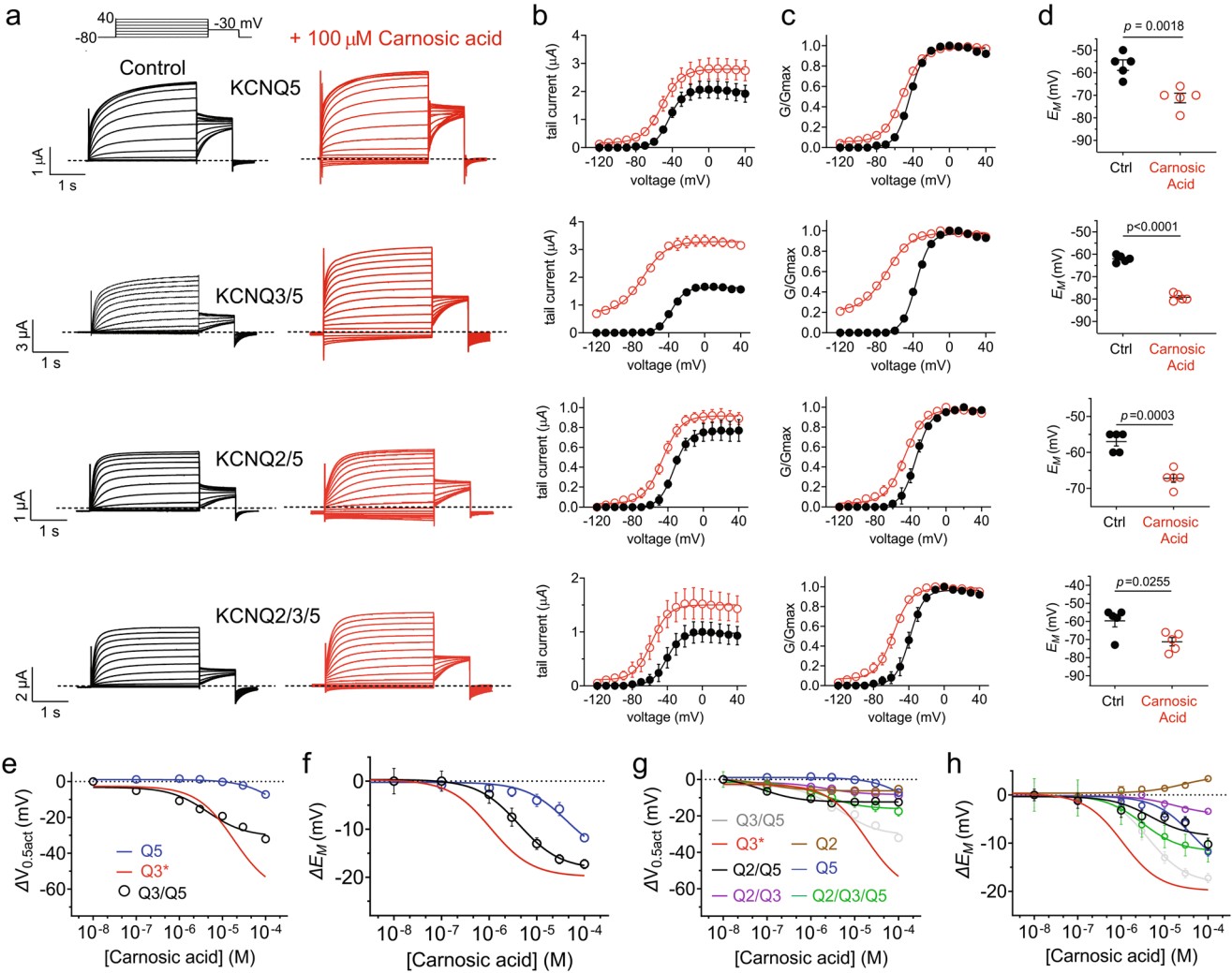

**Fig. 7 Carnosic acid opens heteromeric KCNQ2/5, KCNQ3/5 and KCNQ2/3/5 channels. a** Mean traces for neuronal KCNQ5-containing homomers and heteromers as indicated, expressed in oocytes in the absence (Control) or presence of carnosic acid (100 μM) (red). Scale bars lower left; voltage protocol upper inset; $n = 5$ per group. **b** Mean tail current for channels as in panel a; $n = 5$ per group. **c** Mean normalized tail current (G/Gmax) for channels as in panel a; $n = 5$ per group. **d** Mean unclamped oocyte membrane potential for oocytes expressing channels as in panel A; $n = 5$ per group. **e** Mean $\Delta V_{0.5act}$ versus [carnosic acid] for channels as indicated; $n = 5$ per group; KCNQ3* data from Fig. 4C for comparison. **f**. Mean $\Delta E_M$ versus [carnosic acid] for unclamped oocytes expressing channels as indicated; $n = 5$ per group. KCNQ3* data calculated from Fig. 4D for comparison. **g** Mean $\Delta V_{0.5act}$ versus [carnosic acid] for channels as indicated; $n = 5$ per group; KCNQ2, KCNQ3 and KCNQ2/3* data from Fig. 4 included for comparison. **h** Mean $\Delta E_M$ versus [carnosic acid] for unclamped oocytes expressing channels as indicated; $n = 5$ per group. KCNQ2, KCNQ3* and KCNQ2/3 data from Fig. 4 included for comparison. Error bars indicate SEM. $n$ indicates number of biologically independent oocytes. Statistical comparisons by $t$-test or one-way ANOVA.

headaches from at least as far back as the 13th century, and it is possible that it was grown in Southern England even before the Romans arrived[17,33].

There are so many purported medicinal uses of rosemary, it is easy to be skeptical about efficacy, but a subset of therapeutic effects at least has been supported with preclinical or clinical studies, including neurological benefits in the realm of memory, analgesia, anxiety, depression, epilepsy, alleviation of opium withdrawal symptoms and sleep; and antimicrobial, anti-inflammatory and antioxidant properties[17,34,35]. Rosmarinic acid and carnosic acid are the rosemary phenolics most commonly associated with beneficial effects, largely attributed to their antioxidant and anti-inflammatory actions[34]. Rosemary extract and rosmarinic acid were previously found to inhibit the voltage-gated calcium channel, Cav3.2 (IC₅₀ of 49.9 μM rosmarinic acid). The authors suggested that Cav3.2 inhibition could contribute to the anxiolytic and neuroprotective properties of rosemary[36]. Coupled with our new findings, the combined data suggest that

rosemary may be particularly effective as a neurotherapeutic because of the complementary effects of two of its components (rosmarinic and carnosic acid) on Cav and Kv currents, respectively. Additionally, carnosic acid was previously reported to decrease the binding affinity of [35S]*tert*-butylbicyclopho-sphorothionate to rat brain membranes in vitro, considered to be evidence of binding to a GABA_A receptor, but no functional studies were reported[37]. Carnosic acid is extremely well tolerated in vivo, with acute toxicity studies in mice indicating an oral LD₅₀ of 7.1 g/kg)[38].

The efficacy and potency of carnosic acid (e.g., −25 mV shift in V₀.₅ of KCNQ3/5 activation at 30 μM, and EC₅₀ of 4.53 ± 0.11 μM; Supplementary Table 7) rival those of the synthetic, first-in-class KCNQ channel opening anticonvulsant, retigabine (ezogabine) (−31.8 mV shift in V₀.₅ of KCNQ3/5 activation at 30 μM, and EC₅₀ of 1.4 μM)[20]. Where the two differ is that retigabine, while exhibiting a preference for KCNQ3, is still a highly effective KCNQ2 and especially KCNQ2/3 channel

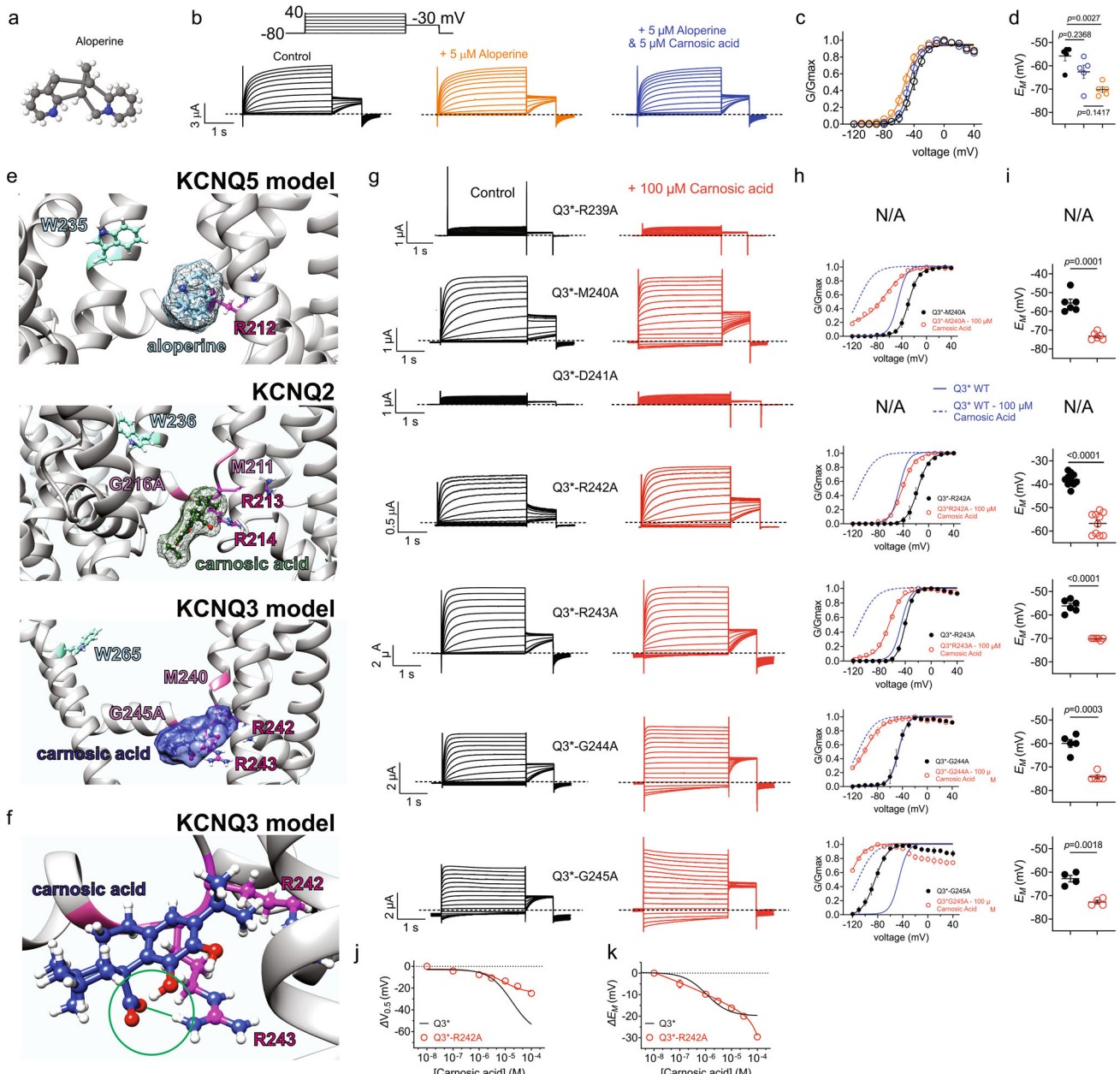

**Fig. 8 Carnosic acid binds to KCNQ3* S4-5 linker arginines. a** Structure of aloperine. **b** Mean traces for KCNQ3/5 expressed in oocytes in the absence (Control) or presence of aloperine and/or carnosic acid (5 μM). Scale bars lower left; voltage protocol upper inset; $n = 5$ per group. **c** Mean normalized tail current (G/Gmax) for KCNQ3/5, pharmacology as in panel B; $n = 5$ per group. **d** Mean unclamped oocyte membrane potential for oocytes expressing KCNQ3/5; pharmacology as in (**b**); $n = 5$ per group. **e** In silico docking results for aloperine in a KCNQ5 model; carnosic acid in KCNQ2 cryo-EM-derived structure[56,57] and in a KCNQ3 AlphaFold model. **f** Close-up of carnosic acid docking to KCNQ3 model structure showing the predicted ionic bond between the carnosic acid carboxyl group and the R243 guanidinium group (green). **g** Mean traces for mutant (as indicated) KCNQ3* expressed in oocytes in the absence (Control) or presence of carnosic acid (100 μM). Scale bars lower left; voltage protocol as in (**b**); $n = 4-10$ per group. **h** Mean normalized tail current (G/Gmax) for KCNQ3* mutants in the absence (black) or presence (red) of carnosic acid; $n = 4-10$ per group. N/A, not applicable. **i** Mean unclamped oocyte membrane potential for oocytes expressing KCNQ3* mutants in the absence (black) or presence (red) of carnosic acid; $n = 4-10$ per group. N/A not applicable. **j** Mean $\Delta V_{0.5act}$ versus [carnosic acid] for R242A KCNQ3*; $n = 4-10$ per group; wild-type KCNQ3* data from Fig. 4 for comparison. **k** Mean $\Delta E_M$ versus [carnosic acid] for R242A KCNQ3*; $n = 4-10$ per group; wild-type KCNQ3* data from Fig. 4 for comparison. Error bars indicate SEM. $n$ indicates number of biologically independent oocytes. Statistical comparisons by one-way ANOVA and paired $t$-test.

opener, and therefore lacks the KCNQ isoform and KCNQ heteromer selectivity of carnosic acid[39]. This endows carnosic acid with unique potential as a probe and potentially therapeutic opener of neuronal KCNQ3/5 channels, over the much more highly studied and better understood KCNQ2/3 channels.

The bactericide triclosan was recently found to activate KCNQ3 but not KCNQ2, also by hyperpolarizing the voltage

dependence of activation, via a predicted binding site at the top of the voltage sensor, similar to what we previously found for quercetin with KCNQ1[40]. However, triclosan also activates KCNQ2/3 channels (EC$_{50}$ of 32 μM)[40]. Triclosan is associated with various health and environmental contamination risks, making it less than ideal as a potential therapeutic KCNQ channel modulator[41], but it remains an additional and potentially useful

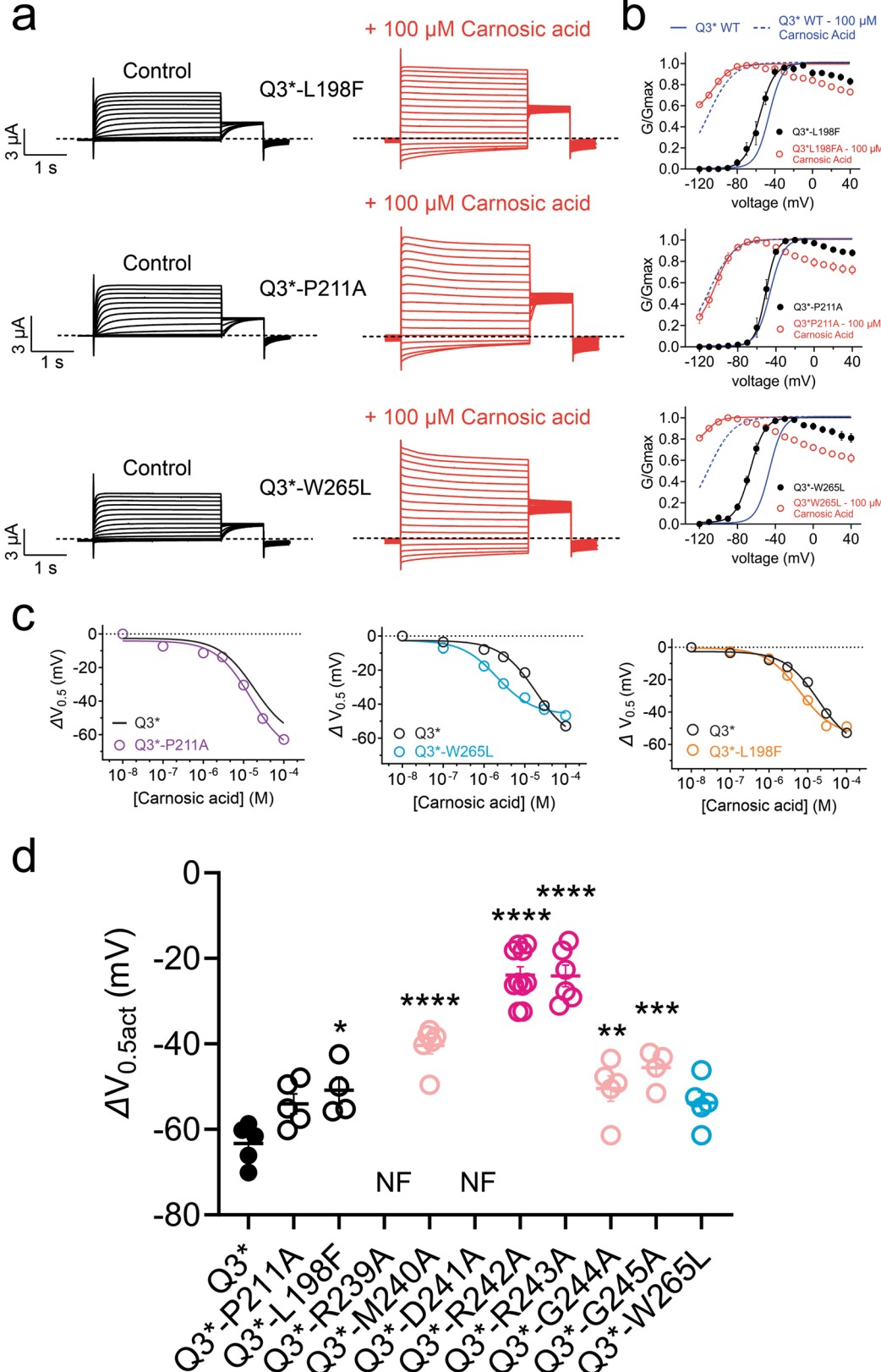

**Fig. 9 Carnosic acid effects are resilient to mutation of previously studied S3 and S5 residues that influence other KCNQ openers. a** Mean traces for wild-type and mutant KCNQ3* channels as indicated in the absence (Control) and presence of carnosic acid (100 μM), $n = 4$ for L198F, 5 for others. **b** Mean normalized tail current (G/Gmax) versus voltage for traces as in (**a**), $n = 4$ for L198F, 5 for others. **c** Mean $\Delta V_{0.5act}$ versus [carnosic acid] for KCNQ3* mutants as indicated; $n = 4$ for L198F, 5 for others; wild-type KCNQ3* data from Fig. 4 for comparison. **d** Mean $\Delta V_{0.5act}$ for wild-type and mutant KCNQ3* channels; $n = 4$ for L198F, 5–10 per group for others; NF nonfunctional. Error bars indicate SEM. $n$ indicates number of biologically independent oocytes. Statistical comparisons by one-way ANOVA. Voltage protocol as in Fig. 8.

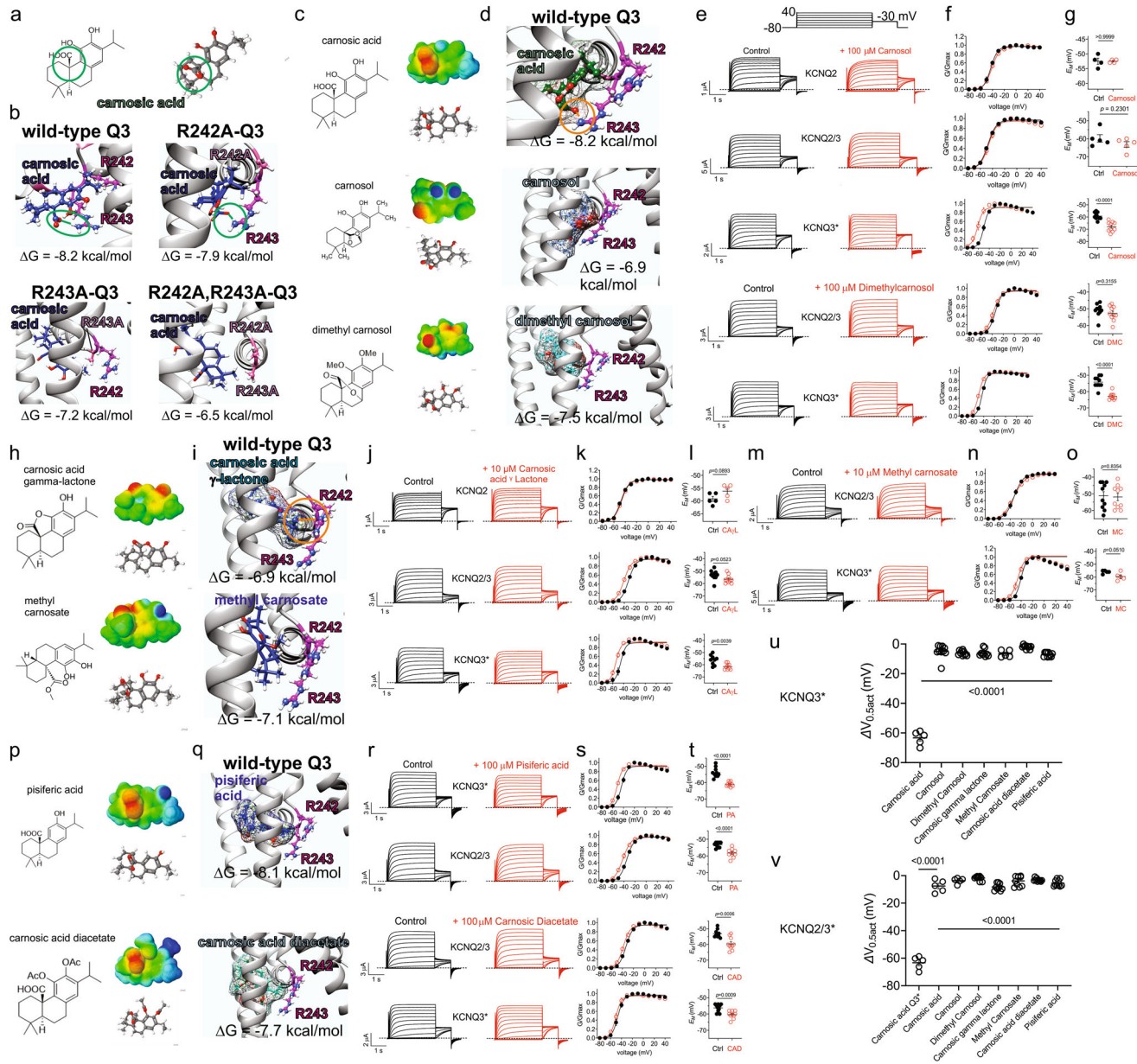

M-current probe, nonetheless. Effects of triclosan on other KCNQ homomers and heteromers have not been described.

Using medicinal chemistry, mutagenesis and in silico docking, we were able to narrow down several features of carnosic acid's KCNQ3* opening action. First, carnosic acid appears almost pre-optimized in this chemical space, i.e., none of a range of chemical modifications made was tolerated vis-à-vis the ability to open KCNQ3*; neither did the modifications make it able to substantially open KCNQ2 or KCNQ2/3. Second, the carnosic acid carboxylate group appears essential for highly efficacious opening of KCNQ3*, via ionic bond formation with the guanidinium group of KCNQ3*-R243. Third, creating an even stronger and more focused electrostatic surface potential at a similar location but using chemistry replacing the carboxylate group with an ester (e.g., methyl ester of carnosic acid) is insufficient to induce robust KCNQ3* opening (Fig. 10). Further medicinal chemistry could be utilized to provide a preferential pharmacokinetic/pharmacodynamic profile for future drug development. It should be noted that the predicted binding free energy (ΔG) values, compared to that of carnosic acid binding to KCNQ3*, were lower for carnosic

acid derivatives binding to KCNQ3* or for carnosic acid binding to KCNQ3*-R242A or KCNA3*-R243A, but not dramatically lower (Fig. 10). Whether this predicted reduction in ΔG reflects a reduction in binding affinity sufficient to explain the reduced functional effects, versus a moderate reduction in binding affinity coupled with a lesser ability of bound carnosic acid and derivatives to actuate channel opening with these mutants, remains to be unambiguously verified. It is already known that carnosic acid crosses the blood brain barrier and is neuroprotective, the latter property previously attributed to activation of antioxidant enzymes via the Nrf2 transcription pathway; carnosic acid has also been discovered to protect against retinal degeneration and oxidative stress[42]. KCNQ5 transcript and protein are found in the retinal pigment epithelium, where it is thought to contribute to the M-current expressed there[43].

We found that, like retigabine and GABA, carnosic acid opening of KCNQ3* is resilient to reduction of PIP$_2$, a lipid-derived signaling molecule that couples voltage sensor activation to channel opening in KCNQ channels[44]. There are at least two interpretations of this result. The first is that carnosic acid

**Fig. 10 Carnosic acid is structurally optimal for KCNQ3 channel opening. a** Structure of carnosic acid highlighting the carboxyl group. **b** In silico docking of carnosic acid to wild-type and mutant KCNQ3 models. **c** Structures and electrostatic surface potentials (red, negative; blue, positive) of carnosic acid and derivatives as indicated. **d** In silico docking of carnosic acid and derivatives to wild-type KCNQ3 model. **e** Mean traces for KCNQ2, KCNQ3* or KCNQ2/3 expressed in oocytes in the absence (Control) or presence of carnosic acid derivatives (100 μM). Scale bars lower left; voltage protocol upper inset; $n = 4$–10 per group. **f** Mean normalized tail current (G/Gmax) for KCNQ2, KCNQ3* or KCNQ2/3 in the absence (black) or presence (red) of compounds as in (**e**); $n = 4$–10 per group. **g** Mean unclamped oocyte membrane potential for oocytes expressing KCNQ2, KCNQ3* or KCNQ2/3 in the absence (black) or presence (red) of compounds as in (**e**); $n = 4$–10 per group; DMC dimethylcarnosol. **h** Structures and electrostatic surface potentials (red, negative; blue, positive) of carnosic acid derivatives as indicated. **i** In silico docking of carnosic acid derivatives to wild-type KCNQ3 model. **j** Mean traces for KCNQ2, KCNQ3* or KCNQ2/3 expressed in oocytes in the absence (control) or presence of carnosic acid γ-lactone (10 μM). Scale bars lower left; voltage protocol as in (**b**); $n = 5$–10 per group. **k** Mean normalized tail current (G/Gmax) for KCNQ2, KCNQ3* or KCNQ2/3 in the absence (black) or presence (red) of carnosic acid γ-lactone (10 μM); $n = 5$–10 per group. **l** Mean unclamped oocyte membrane potential for oocytes expressing KCNQ2, KCNQ3* or KCNQ2/3 in the absence (black) or presence (red) of carnosic acid γ-lactone (CAγL) (10 μM); $n = 5$–10 per group. **m** Mean traces for KCNQ3* or KCNQ2/3 expressed in oocytes in the absence (Control) or presence of methyl carnosate (100 μM). Scale bars lower left; voltage protocol as in panel e; $n = 5$–10 per group. **n** Mean normalized tail current (G/Gmax) for KCNQ3* or KCNQ2/3 in the absence (black) or presence (red) of methyl carnosate (100 μM); $n = 5$–10 per group. **o** Mean unclamped oocyte membrane potential for oocytes expressing KCNQ3* or KCNQ2/3 in the absence (black) or presence (red) of methyl carnosate (100 μM); $n = 5$–10 per group. MC methyl carnosate. **p** Structures and electrostatic surface potentials (red, negative; blue, positive) of carnosic acid derivatives as indicated. **q** In silico docking of carnosic acid derivatives to wild-type KCNQ3 model. **r** Mean traces for KCNQ2, KCNQ3* or KCNQ2/3 expressed in oocytes in the absence (Control) or presence of carnosic acid derivatives (10–100 μM). Scale bars lower left; voltage protocol upper inset; $n = 10$ per group. **s** Mean normalized tail current (G/Gmax) for KCNQ2, KCNQ3* or KCNQ2/3 in the absence (black) or presence (red) of compounds as in (**r**); $n = 10$ per group. **t** Mean unclamped oocyte membrane potential for oocytes expressing KCNQ2, KCNQ3* or KCNQ2/3 in the absence (black) or presence (red) of compounds as in (**r**); $n = 10$ per group; CAD Carnosic acid diacetate; PA pisiferic acid. **u** Mean $\Delta V_{0.5act}$ for KCNQ3* in response to carnosic acid and derivatives; $n = 5$–10 per group. **v** Mean $\Delta V_{0.5act}$ for KCNQ2/3 in response to carnosic acid and derivatives; $n = 5$–10 per group. KCNQ3* response to carnosic acid (grey; from (**u**)) shown for comparison. Error bars indicate SEM. $n$ indicates number of biologically independent oocytes. Statistical comparisons by one-way ANOVA.

replaces or obviates the need for $PIP_2$, and that regardless of the presence of $PIP_2$, carnosic acid maintains its opening ability. The alternative explanation is that carnosic acid stabilizes $PIP_2$ binding to the channel but may still require $PIP_2$ for carnosic acid to open KCNQ3, as previously suggested for retigabine[22]. Here, with wortmannin pretreatment to reduce $PIP_2$ levels prior to carnosic acid addition, the efficacy of carnosic acid at its KCNQ3* EC_{50} (5 μM) was reduced by a statistically significant but moderate degree, not unequivocally supporting either model. Carnosic acid (5 μM) was able to on average prevent KCNQ3* current decay over 15 s by activation of DrVSP at +120 mV to deplete $PIP_2$, which in the absence of carnosic acid resulted in 50% current loss. Again, this does not support either model unequivocally, and we do not yet have sufficient evidence to favor either model over the other. There is a precedent for a $PIP_2$ replacement model, i.e., CP1, the synthetic, sulfate-containing $PIP_2$ replacement discovered by in silico docking screening to KCNQ1[45]. Interestingly, much like $PIP_2$, carnosic acid is an amphipathic molecule - carnosic acid contains a polar carboxylic acid and phenol groups and a non-polar diterpene framework. We consider it likely that its affinity for KCNQ3 is a function of both ionic, hydrogen-bond and hydrophobic interactions.

Intriguingly, nothing in the predicted KCNQ3 binding region of carnosic acid hints at the mechanism for its selectivity, and unusually, the KCNQ3*-W265L mutant is at least as sensitive as KCNQ3* to carnosic acid, unlike retigabine and GABA. We interpret these findings as evidence that carnosic acid, like CP1, binds deep in the binding pocket between the pore and VSD, unlike retigabine and GABA, which bind closer to W265[26,30,31]. Unlike CP1[45], carnosic acid does not open homomeric KCNQ1. In terms of $V_{0.5act}$ hyperpolarization, CP1 is most active on KCNQ1/KCNE1, followed by KCNQ1 and KCNQ3* then KCNQ2[26,30,31] (KCNQ5 data not reported); carnosic acid is most active on KCNQ3*, followed by KCNQ5, with negligible effects on KCNQ1 or KCNQ2. Similar to GABA, which we found physically binds to KCNQ2–5 (via the S5 Trp, W265 in human KCNQ3) but only activates KCNQ3* and KCNQ5[26], our current data are consistent with carnosic acid selectivity being functional (i.e., binding opens some KCNQ isoforms but not others) rather

than arising from its ability to bind per se. Further studies are needed to understand what imparts this functional selectivity.

Interestingly, it was previously reported that while the presence of only one retigabine-sensitive subunit is needed for near-maximal retigabine sensitivity, effects of ICA73, which is thought to interact with the KCNQ3* voltage sensor, are highly sensitive to mutation of even one subunit in the tetramer[46]. This sensitivity might be what gives greater KCNQ isoform selectivity to voltage sensor domain-targeted small molecules versus the less KCNQ-selective retigabine. As to the precise molecular correlate of that selectivity, it does not necessarily need to be in the VSD; as discussed above, while binding may occur in the VSD, transduction of that binding into enhanced activation could depend on other segments. The largest deviation between KCNQ3 and other neuronal isoforms lies between S5 and the selectivity filter, where KCNQ3 contains ten extra residues, but there are also sporadic sequence differences throughout the transmembrane segments, for future study[47].

It is important to note that a limitation of the study is that we did not verify the relative expression of the protein of each KCNQ isoform when co-expressing multiple subunits. Oocytes are superior to mammalian cells for multi-subunit transient expression studies as each oocyte is injected independently with cRNA for each subunit, versus transfection of a population of mammalian cells, which is more likely to result in higher variability of the relative proportion of subunits in a given cell. Nevertheless, it is possible that there was variability in the relative isoform expression in a given oocyte, and that may have contributed to modest variability in the midpoint voltage dependence of activation and activation rate between control groups for a given subunit combination. Other sources of variability could also be temperature fluctuation, or differences in $PIP_2$ levels or other oocyte factors between different groups, as the recordings were conducted over many months. We do not consider this a significant limitation as each oocyte acts as its own baseline control for comparison to properties after addition of extract or compound, and in our experience, voltage shifts attributable to a specific small molecule and specific channel isoform are similar despite modest changes in the starting midpoint voltage dependence.

Carnosic acid opens up novel experimental and potential therapeutic avenues due to its unique combination of high efficacy and isoform selectivity, combined with a low-micromolar potency that brings it into the useful range for either type of application. From an experimental standpoint, having a small molecule that can readily distinguish between KCNQ3/5 over KCNQ2/3 should prove invaluable in delineating the role of either complex in neuronal and other cell types, and with the correct dosage and voltage protocol, one could even use carnosic acid to distinguish KCNQ3/5 from KCNQ2/5 and KCNQ2/3/5, the latter two complexes being very recently discovered in mouse brain[4]. Specifically, while carnosic acid also opens heteromeric KCNQ2/5 and KCNQ2/3/5 channels, also thought to form in neurons[4], if one holds the cell at −120 mV, carnosic acid is 100% selective for KCNQ3/5 over the other known neuronal heteromers (Fig. 7). From a therapeutic standpoint, as the various roles of the neuronal KCNQ heteromers are teased out, there may be clinical opportunities for a KCNQ3/5-selective opener that we do not yet understand – perhaps guided by ancient uses of rosemary as an herbal medicine. Our findings further reinforce the enormous potential of plants to continue to deliver small molecules with novel activities, and coupled with this the critical need to acknowledge, learn from and preserve traditional botanical medicine practices and the natural environments that support them.

## Methods

**Collection and preparation of plant extracts.** *Salvia rosmarinus* samples (flowers, leaves, and entire aerial parts) were collected from the corresponding author's garden by his children and then frozen until the day of extraction. We homogenized the samples using a bead mill with porcelain beads in batches in 50 ml tubes (Omni International, Kennesaw, GA, United States). We resuspended the homogenates in 80% methanol/20% water (100 ml per 5 g solid) and then incubated for 48 h at room temperature, occasionally inverting the bottles to resuspend the extracts. We next filtered the extracts through Whatman filter paper #1 (Whatman, Maidstone, UK), and then removed the methanol using evaporation in a fume hood for 24–48 h at room temperature. We centrifuged extracts for 10 min at 15 °C, 4000 RCF to remove the remaining particulate matter, followed by storage at −20 °C. On the day of electrophysiological recording, we thawed the extracts and diluted them 1:100 in bath solution (see below) immediately before use.

**Channel subunit cRNA preparation and Xenopus laevis oocyte injection.** We generated cRNA transcripts encoding human KCNQ1, 2, 3, and 5 (wild-type and mutant) by in vitro transcription using the mMessage mMachine kit (Thermo Fisher Scientific, Waltham, MA, USA) according to manufacturer's instructions, after vector linearization, from cDNA sub-cloned into expression vectors (pTLNx and pXOOM) incorporating *Xenopus laevis* β-globin 5′ and 3′ UTRs flanking the coding region to enhance translation and cRNA stability. Mutant KCNQ cDNAs were generated by Genscript (Piscataway, NJ, USA) and cRNAs made as above. We injected defolliculated stage V and VI *Xenopus laevis* oocytes (Xenoocyte, Dexter, MI, USA) with KCNQ cRNAs (2–10 ng) and incubated the oocytes at 16 °C in ND96 oocyte storage solution containing penicillin and streptomycin, with daily washing, for 2–4 days prior to two-electrode voltage-clamp (TEVC) recording.

**Two-electrode voltage clamp (TEVC).** We performed TEVC at room temperature using an OC-725C amplifier (Warner Instruments, Hamden, CT, USA) and pClamp10 software (Molecular Devices, Sunnyvale, CA, USA) 2–4 days after cRNA injection. We placed oocytes in a small-volume oocyte bath (Warner) and viewed them with a dissection microscope for cellular electrophysiology. We sourced chemicals from Sigma-Aldrich (St. Louis, MO, USA) and Combi-Blocks (San Diego, CA, USA). We studied effects of *Salvia rosmarinus* extracts and of compounds, solubilized directly in bath solution (in mM): 96 NaCl, 4 KCl, 1 MgCl$_2$, 1 CaCl$_2$, 10 HEPES (pH 7.6). We introduced extracts or compounds into the oocyte recording bath by gravity perfusion at a constant flow of 1 ml per minute for 3 min prior to recording. Pipettes were of 1-2 MΩ resistance when filled with 3 M KCl. We recorded currents in response to voltage pulses between −120 mV and +40 mV at 10 mV intervals from a holding potential of −80 mV, to yield current-voltage relationships and examine activation kinetics. We analyzed data using Clampfit (Molecular Devices) and Graphpad Prism software (GraphPad, San Diego, CA, USA), stating values as mean ± SEM. We plotted raw or normalized tail currents versus prepulse voltage and fitted with a single Boltzmann function:

$$g = \frac{(A_1 - A_2)}{\left\{1 + \exp[V_{\frac{1}{2}} - V/Vs]\right\} y + A_2} \qquad (1)$$

where $g$ is the normalized tail conductance, $A_1$ is the initial value at -∞, $A_2$ is the final value at +∞, $V_{1/2}$ is the half-maximal voltage of activation and $V_s$ the slope factor. We fitted activation and deactivation kinetics with single exponential functions.

**Chemical synthesis.** Carnosic acid was obtained from Combi-Blocks (San Diego, CA.) or Sigma-Aldrich (St. Louis, MO). Pisiferic acid was obtained from Combi-Blocks. Carnosol was commercially available from Sigma-Aldrich or prepared from carnosic acid using the method of Han et al.[48]. Carnosic acid diacetate, methyl carnosate and carnosic acid γ-lactone were prepared using the method of Han, et al.[48]. Conc. hydrochloric acid (35–38% in water) was from Fisher. Methanol (MeOH), ethyl acetate (EtOAc), toluene, dichloromethane (CH$_2$Cl$_2$) and hexanes were HPLC or LC/MS grade from Fisher or VWR International. Anhydrous acetone was supplied by Sigma-Aldrich. Water was 18.2 mΩ-cm from a Barnstead NANOpure Diamond$^{TM}$ system. Trifluoroacetic acid was obtained from EMD Millipore. Melting points (mp) were determined on an Electrothermal MEL-TEMP 3.0 apparatus (Barnstead International, Dubuque, IA) and are not corrected.

Preparative HPLC separations were carried out using a Shimadzu system consisting of two LC-8A pumps, a fraction collector (FRC-10A), a SIL-10AP auto sampler, a diode array detector (CPD-M20A) and a CBM-20A communication module. The separations employed a Waters PREP Nova-Pak® HR C18 6 μM 60 Å 40 ×100 mm reversed phase column with a 40 ×10 mm Guard-Pak insert and a Waters PrepLC Universal Base. The solvent system employed was MeOH/water gradients both containing 0.1 % TFA. Fractions were collected based on their response at 254 nm. Flash chromatography was carried out on 230–400 mesh silica gel (Silica gel 60, Geduran) obtained from Fisher Scientific using the method of Still et al.[49]. Columns were eluted with EtOAc/hexanes mixtures or CH$_2$Cl$_2$. Thin layer chromatography (TLC) employed Analtech GHLF UV254 Uniplate$^{TM}$ silica gel plates from Miles Scientific. Plates were visualized under UV light or with I$_2$ vapor.

Mass spectroscopy employed a Thermo Scientific TSQ Quantum Ultra triple-stage quadrupole mass spectrometer. Heated-electrospray ionization (H-ESI) was used in negative or positive ionization mode depending on the structure of the analyte. Automatic methods for the optimization of instrument parameters were used to maximize sensitivity. Samples were analyzed by direct injection in MeOH or MeOH/water (TFA conc kept at 0.01% or less) using a syringe pump.

For carnosic acid diacetate synthesis, carnosic acid (1.02 g, 3.05 mmol) in CH$_2$Cl$_2$ (13 mL) was treated with neat acetic anhydride (1.6 mL) and 4-dimethylaminopyridine (DMAP, 443 mg). The reaction was warmed after the DMAP was added. The resulting yellow solution was stirred at rt for 2 days. The reaction was then diluted with CHCl$_3$ (40 mL) and extracted with a 1 M aq. HCl solution (25 mL). The organic layer was separated and washed with brine, dried (MgSO$_4$), filtered, and conc to dryness. The crude product was purified by flash chromatography with a gradient from 4:1 to 3:1 hexanes/EtOAc. The product was isolated as an oil (480 mg) that solidified on standing. The resulting solid had mp 94–95 °C (lit mp 158–159 °C[50]. MS ESI⁻ m/z 415 (M – H⁺).

For methyl carnosate synthesis, carnosic acid (329 mg, 0.99 mmol) was dissolved in 5:1 toluene/MeOH (6 mL) under N$_2$ and cooled in an ice-water bath. The reaction was treated with a 1.8 to 2.4 M solution of TMSCN$_2$ in hexanes (Thermo-Fisher; 1.1 mL) added dropwise via syringe. The reaction was stirred cold and then allowed to warm to rt and stir overnight. The reaction was cooled in an ice-water bath and a 2 M aq. HOAc solution was added (10 mL). The two-phase mixture was partitioned between water and EtOAc. The organic layer was separated, washed with brine, dried (MgSO$_4$), filtered, and conc. in vacuo. Flash chromatography with 9:1 hexanes/EtOAc gave 191 mg of the methyl ester as a white solid with mp softens at 118 °C, melts 126–127 °C (lit mp 118–119 °C[51]. MS ESI⁻ m/z 345 (M – H⁺).

For carnosic acid γ-lactone synthesis, carnosic acid (125 mg, 0.30 mmol) in 3 mL of CH$_2$Cl$_2$ was treated with solid DCC (86 mg). A ppt formed almost immediately. Sold DMAP (7.7 mg) was then added. The mixture was stirred at rt under N$_2$ for 5 h and then filtered. The mother liquor was conc to dryness and purified by flash chromatography (100% CH$_2$Cl$_2$). The lactone (40 mg) was isolated as a yellow foam. Lit mp for the lactone is 106–109 °C[52]. MS ESI⁻ m/z 313 (M – H⁺).

The dimethyl ether of carnosol was prepared using the method of Luis et al. (Luis, J. G. et al.[53]. except that the compound was purified by reverse-phase HPLC.

For di-d$_3$-methyl carnosol synthesis, carnosol (100 mg, 0.30 mmol) in acetone (16 mL) in an ice-water bath was treated with neat d$_3$-MeI (500 μL, 1.16 g, 8.00 mmol) and 2 eq. of K$_2$CO$_3$ (84 mg, 0.60 mmol). The resulting mixture was stirred cold and then allowed to warm to rt and stir overnight. The reaction was diluted with cold water and extracted with EtOAc (3 × 40 mL). The pooled organic layer (yellow) washed with brine, dried (MgSO$_4$), filtered and conc to dryness, tare 130.835 g, weight 138 mg. Crude product was purified by RPHPLC (20 to 100% MeOH/water containing 0.1% TFA). Fractions with RT 6.6 min were collected and concentrated to 13 mg of di-d$_3$-methyl carnosol as an oil; lit mp 155–157 °C[50]. MS (ESI⁺) m/z 365 (M + H⁺) and 387 (M + Na⁺).

**In silico docking**. We plotted and viewed chemical structures and electrostatic surface potential using Jmol, an open-source Java viewer for chemical structures in 3D: http://jmol.org/. For in silico ligand docking predictions of interaction with KCNQ channels, we performed unguided docking to predict potential interaction sites, using SwissDock with CHARMM forcefields[54,55], the cryo-EM-derived KCNQ2 structure (PDB: 7CR0)[27], the AlphaFold[28,29]-predicted KCNQ3 model structure (Supplementary Structure File 1) and the neuronal KCNQ (KCNQ5) model structure (Supplementary Structure File 2)[15].

**Statistics and reproducibility**. All values are expressed as mean ± SEM. Multiple comparison statistics were conducted using a One-way ANOVA with a post-hoc Tukey HSD. Comparison of two groups was conducted using a $t$-test; all $p$ values were two-sided. Electrophysiological data were confirmed in at least two batches of oocytes. Biological replicates are defined as numbers of oocytes; sample sizes are given in the figure legends.

**Reporting summary**. Further information on research design is available in the Nature Portfolio Reporting Summary linked to this article.

## Data availability

Source data for Figs. 1–10 are available at Dryad data repository: https://doi.org/10.7280/D1GH5Q.

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

## Acknowledgements

This study was supported by the National Institutes of Health, National Institute of Neurological Disorders and Stroke (NS107671) to G.W.A. and a Susan Samueli Integrative Health Institute, Samueli Scholarship to GWA. We are grateful to Dr. Ryan Yoshimura for technical assistance and advice, to George Abbott and Victoria Abbott for collecting *Salvia rosmarinus* samples, and to Bo Abbott for *Salvia rosmarinus* images.

## Author contributions

G.W.A. and R.W.M. conceived the study; R.W.M. conducted TEVC analysis and analyzed data; D.H. conceptualized, synthesized, and analyzed chemicals, G.W.A. oversaw and obtained funding for the project; R.W.M., D.H., and G.W.A. prepared the figures, G.W.A. wrote the manuscript; all authors edited the manuscript.

## Competing interests

The authors declare no competing interests.
