## [Peer Review File · Communications Biology]

Reviewers' comments:

Reviewer #1 (Remarks to the Author):

In this paper, the authors identify and characterize a novel chemical activator for selected members of the KCNQ/Kv7 channel family. The conclusions of the authors are overall well supported by a large amount of electrophysiology recordings, supplemented by a thorough mechanistic investigation using *in silico* studies, mutagenesis, and chemical synthesis. This contribution represents a significant discovery that will help move the field forward. The manuscript would be much improved after fixing several inconsistencies in data interpretation as well as few minor points described below:

Major points:

A "channel opener" effect logically exists if the stimulus alone is sufficient to significantly increase the channel macroscopic conductance above resting level. This seems to be the case with RAP against KCNQ3/5 (Fig 2I) and with CA against Q3* (Fig 3C, Fig 4) and KCNQ3/5 (Fig 7B). However, the weaker effect of RAP on Q1, Q2, Q3* and Q2/3 and the weaker effect of CA on Q5, Q2/5 and Q2/3/5 (Figure 7) does not justify, in my opinion, using the word "opener" to describe these effects, simply because conductance remains near zero at negative voltages. The authors should use different words to highlight the distinction between "opener" vs. weaker activatory effects.

Page 7, 1st paragraph: The expression "under PiP2 replete conditions" is incorrect. There are only two conditions in Fig 5D: with Wortmannin (PiP2 depletion) or without Wortmannin (no change in PiP2). Using "replete" suggests PiP2 has been depleted first and then added back to the original level. This is not what happened according to the figure.

Figure 8: The mutant phenotypes shown in Figure 8G-H should be compared with WT KCNQ3 data Figure 3A. Could the authors add reference G/Gmax curves (in both presence and absence of CA) to the G/Gmax plots in figure 8H? In addition, no discussion of phenotypes for other mutants (R239, M240, D241, G244) is given. I understand these mutations do not significantly alter the CA effects but this needs to be quantified or at least discussed.

Figure 9: Same as Figure 8: it would be nice to show reference G/Gmax curves for WT KCNQ3* to better visualize mutant phenotypes without going back and forth between figures.

Page 9, 1st paragraph: The EC50 plot in Figure 8J (right panel) does not show a full sigmoid curve for the Q3*-R242A mutant and thus cannot be unambiguously interpreted. It seems that the red line is a fit to the data using some equation. If that is the case, the equation used for EC50 determination must be stated somewhere in the manuscript. If EC50 for CA is 0.1mM for this mutant (lower estimated boundary given by the authors), a symmetrical extrapolation of this fitted curve for saturation concentration of CA would give an Em shift of at least -60 mV (i.e. two times the -30mV shift observed at the EC50 value). This change would move Em from about -40 mV (control oocytes) to about -100 mV, which is below the K+ equilibrium potential (about -80 mV according to the reversal potential of K currents seen in most current traces). Because this theoretical value cannot be attained, the extrapolation of the fitted sigmoid curve cannot be used to estimate EC50. A second, more concerning problem with the estimated 100-fold decrease of EC50 is that it does not agree with the delta-V0.5 data from the left panel in the same figure: while the left panel shows a 2-fold decrease in efficacy (as correctly stated by the authors), there is little to no visible change in EC50. In my opinion, the data suggests that the mutation decreases maximal CA efficacy with no change in EC50.

Page 9, 2nd paragraph: "ten-fold more sensitive in term of potency" seems exaggerated. The middle panel in Figure 9C shows that the blue curve shifts by about 2/3 third of a decade in the log x-axis. A

tenfold effect would have required a shift by a full decade. A 2/3 decade leftward shift corresponds to a 5-fold increase in EC50.

Figure 10: In silico data suggests arginine mutations reduces binding but only weakly. There is no statistical error for the free energy calculations. Thus it is difficult to interpret this data. Regardless, the binding of CA remains strong in both WT and mutants. This data suggests that the mutants' phenotype does not come from a decrease in binding free energy, as stated in the text. Can the authors comment on this?

Minor points:

Fig 4H, x-axis label: "-120 mV" label in the voltage axis is missing (it is moved to panel G)

Fig 5F, y axis title: "current decay" (not "deacy")

Page 8, 2nd paragraph: Figure 8 A-C shows data from KCNQ5, not KCNQ3/5 as stated in the text.

Page 11, 2nd paragraph: A negative sign should be put in front of binding free energies values

Page 14: I do not understand the clause: "..., and that both channel-bound and unbound PiP2 is depleted, without significant effects on CA opening ability". The verb "is" refers to a singular subject but in this case, the subject "both channel bound- and unbound PiP2" should be plural. Do the authors mean to say something like: "..., and that regardless of the presence of PiP2, CA maintain its opening ability"? Please clarify this sentence.

Page 15, 1st paragraph: The concluding sentence "...CA selectivity being functional rather than binding selectivity, i.e. in the effects of CS binding rather than the binding itself." is difficult to parse. Grammatically, "rather than" in the first clause should be followed by an adjective (to oppose "functional"), not a noun ("binding selectivity"). The second explanatory clause does not help much. I get what the authors want to say in this sentence but only after deciphering its meaning. Rephrasing would make the idea clearer.

Page 15, 2nd paragraph: "holding the cell at -120mV": this voltage is more negative than what a neurons can achieve in normal physiological conditions. This sentence does not align well within the rest of the paragraph dealing with in vivo/clinical applications.

Discussion: It would be nice if the authors could discuss hypothetical mechanisms by which channels with different subunit stoichiometry end up having different response to CA.

Jerome Lacroix

Reviewer #2 (Remarks to the Author):

The present work " Ancient medicinal plant rosemary contains a highly efficacious and isoform selective KCNQ potassium channel opener" is an interesting study that aims to characterize the effect of natural plant extracts on neuronal KCNQ channels with therapeutic potential. Using electrophysiology, mutagenesis, and in silico docking the authors showed that Carnosic acid (CA) – mainly– activates neuronal KCNQ3 and KCNQ3/5 channels over other KCNQ homo- and heteromeric complexes, somewhat underscoring CA-subunit selectivity. Overall, the experiments are nicely done

and yield sound data that support the major claims of the paper. The manuscript is well structured and well written, and figures are well presented. The study proposes putative interaction sites of CA with KCNQ channels, which could be valuable to develop potential new neuroprotective drugs targeting KCNQ channelopathies. There are few suggestions that would further strengthen the report and a few other areas in need of further clarifications.

Major comments:

1) Page 7: "Here, depletion of PIP2 using pretreatment with wortmannin reduced KCNQ3* current magnitude ~fivefold, yet carnosic acid 'CA' (5 μ M) was still able to shift the KCNQ3* $V_{0.5act}$ (Figure 5A, B) and EM (Figure 5C) under these conditions, albeit the $\Delta V_{0.5act}$ was ~50% of that under PIP2 replete conditions (Figure 5D)".

This sentence, and the data itself, is confusing and needs to be fixed. PIP2 is either depleted by Wortmannin and therefore KCNQ3 channel cannot open, which is not what the black-control trace in Fig 5A bottom panel shows (it shows only a reduction), or alternatively, Wortmannin only partially reduces PIP2 levels such that there is still a fraction of active KCNQ3 channels that can respond to CA. The conclusion that carnosic acid (5 μ M) can shift the KCNQ3* $V_{0.5act}$ even if PIP2 is 'depleted' needs to be revisited. The argument that Wortmannin reduces the current by 5-fold does not justify PIP2 'depletion' whatsoever.

2) CA structure resembles more (although not quite) that of retigabine and ICA-069673 compounds than that of PIP2. How would CA and PIP2 compete for the same spot in the S4-S4-S5 loop? I would envision, much like for PIP2, that a drug acting in the proposed S4-S4-S5 linker interphase would behave like an amphipathic molecule. Is this the case of CA? This is not commented in the discussion. The authors did comment, however, in the discussion (page 14) that, "...carnosic acid does not require PIP2 and may even compete with PIP2 and replace it under some circumstances." This is far too speculative...a suggestion simply lacking enough experimental support.

3) Along this line, the authors put forward the idea of CA interacting with the same PIP2-binding residues in the S4-S5 linker. Yet, PIP2 is believed to bind this pocket (but also the cytoplasmic ends of the VSD, the gate, helices A and B and their linkers, at least in homologous KCNQ2, 4 channels) in a state-dependent manner (only in the activated state, PMID: 24277843, PMID: 34650221, PMID: 29078287). Is CA only interacting with the S4-S5 linker and/or with all these other hotspots?... in a state dependent fashion? I think that the data provided in the current work is insufficient to formulate a precise picture of the 'binding' (maybe re-wording it to interacting sites, or effect?... See also below) site of CA onto KCNQ3 channels and more experiments need to be done to support such claim.

4) Page 9, the authors state: "Overall, the data strongly suggest that KCNQ3 R242 and R243 are the most important components of the carnosic acid binding site, and likely directly coordinate carnosic acid binding". I recommend rephrasing this conclusion and reduce the tone of the claim. While the data in Figure 8 shows that Q3-R242A and R243A reduced the CA-mediated activation, especially when compared with the small effect shown by residues L198F, P211A, and W265L in Fig. 10, according to the tables of Figures 8-data, 100 μ M of carnosic acid still induced a GV shift of ~24 mV and a strong negative shift in Em in both Q3-R242A and Q3-R243A. While the data shown here suggests that R242 and R243 might participate in the CA activating effect, the evidence provided do not show that these residues are component of the binding site, much less that these residues directly coordinate CA binding. The authors convincingly show that CA effect on KCNQ3 is not through the retigabine and ICA/73 'binding' sites, which is already an important finding.

5) Overall, the control traces for KCNQ3* (and KCNQ2/3, please compare Figs 2, g and 10) are not

consistent throughout the figures, which raises concerns about the expression of the intended channel/subunit. For example, Fig. 10E, J, and R, looking at the traces denoted as KCNQ3* and Q3* in black (the nomenclature of KCNQ3* or Q3* ...should be also consistent) they all showed different kinetic of activation, particularly striking in the tail current. The same inconsistency is true for the G/Gmax V0.5 and Em values, which are all too variable, spanning a V-range from -50 to -40 mV in the V0.5 and -50 to -60 mV for the Em, in the example showed in carnosol and dimethylcarnosol, for instance. The same is true for instance in Figures 3C and 5B, G/Gmax, the V0.5 before Rosmarinic acid and before Wortmannin are -53 mV and -32 mV, respectively, which to me is beyond the permissive variable range. Furthermore, simple inspection of the GV curves (and V0.5) of KCNQ3* control (black trace) before carnosol in Fig.10F and methyl carnosate in Fig.10N, third and second graph from top, respectively, showed very different GV curves compared to similar control GV curves (e.g. for those before dimethylcarnosol, or any of the GVs shown for KCNQ3* in the entire paper). This is important because for instance, assuming that after correcting the (almost certain wrong) weird GV curve in Fig.10F the V0.5 is back to ~ 40mV (as shown throughout), then carnosol would then have induced a strong ~ 20mV hyperpolarizing shift of the GV, which would change the authors claims. These issues, and the concomitant conclusion derived from them need to be thoroughly revised throughout the ms. and an explanation needs to be provided.

6) The authors state in the discussion: "Rosmarinic acid and carnosic acid are the rosemary phenolics most commonly associated with beneficial effects, largely attributed to their antioxidant and anti-inflammatory effects³⁴, but regulation of ion channels was not previously reported." Yet, PMID: 29073181, for instance, reported that Rosemary extracts inhibited T-type Ca²⁺ (CaV3.2) channel currents and left-shifted the steady-state inactivation of CaV3.2. The authors should take advantage of this important report and discuss the synergistic effect of Rosemary extract on (activating) KCNQ and (inhibiting) Ca3.2 channels, particularly because Cav3.2, much like KCNQ2/3, contributes to control neuronal excitability.

7) In Fig. 7: what was the injected Q2/Q3/Q5 ratio? If it was 1:1:1, what is the composition of the 'four' missing subunit in the tetramer?

Minor comments

1) Fig.3 legend: "Mean unclamped oocyte membrane potential for KCNQ2-expressing oocytes as" This is wrong, right? Should not it be KCNQ3?

2) Furthermore, in Figures 3B, 6C, and 6F, Hesperidin: the black plot shows the peak-current/V but not the tail-current/V curve under control conditions, like for the other conditions...why?

3) Page 6: "...KCNQ2 insensitivity to carnosic acid is dominant in KCNQ2/3 heteromers". Figure 4J and S-Fig4-Table show that carnosic Acid cause around -10 mV shift of the G(V), so I would reword this here and maybe say instead "...KCNQ2 reduced sensitivity to carnosic acid" or something similar.

4) Along this line on Page 7, "...yet hesperidin did not open KCNQ2 (Figure 6D-F)". It should be referred to Fig.6 E-G, right? Please, fix it.

5) Fig 5F, change deacy to decay.

Reviewer #3 (Remarks to the Author):

Thank you for the invitation to review "Ancient medicinal plant rosemary contains a highly efficacious and isoform selective KCNQ potassium channel opener" by Manville and colleagues. This is an interesting and comprehensive paper that describes the identification of carnosic acid in rosemary extract as a powerful activator of KCNQ3 subunits, with intermediate effects on KCNQ5, but minimal effects on KCNQ2. This leads to interesting selectivity towards different KCNQ heteromers, with relatively weak effects on Q2/Q3 heteromers, but strong potentiation of Q3/Q5 heteromers. Using mutagenesis, docking, and testing of chemical analogs of carnosic acid, the authors support a model of this compound binding in the vicinity of an arginine doublet at the intracellular end of the S4 segment. Overall, this study contains a significant amount of data and is quite convincing. I only have a few suggestions/questions for clarification.

Page 7 - 'dominant negative effects of KCNQ2' was a little confusing to me at first - perhaps describe as 'the insensitivity of KCNQ2 that dominated the response of KCNQ2/3 heteromers' or something like that.

Page 8... 'Thus carnosic acid likely underlies the KCNQ3/5 opening ability of heteromeric KCNQ3/5 channels.' I didn't understand the wording of this sentence, just a typo or something missed in the draft.

Page 8... 'Interestingly, the in silico docking predicted ionic bond formation between carnosic acid via its carboxylate group to the guanidinium group of R243 in KCNQ3, but not in KCNQ2'. Since these residues are present in KCNQ2,3,4,5, perhaps the authors could clarify whether there are other contributors in this region that are candidates for the apparent selectivity? If there is nothing obvious, that's ok too as the functional outcome is so clear, but perhaps worth commenting.

Page 15... "our current data are consistent with carnosic acid selectivity being functional rather than binding selectivity, i.e., in the effects of carnosic acid binding rather than the binding itself". Related to the previous comment/question - I was unclear what the authors were getting at here, so perhaps clarify. My impression earlier was that there was not predicted binding/interaction of carnosic acid to Q2. However, is this sentence implying binding with no effect?

Overall this is an interesting study that will extend our understanding of KCNQ channel modulation, and highlights ways to modulate specific subtype combinations.

We are extremely grateful to the reviewers for their thorough and highly positive reviews. We describe our responses point-by-point below in bold (page and paragraph numbers correspond to the “track changes” version of the revised manuscript).

Reviewer #1 (Remarks to the Author):

In this paper, the authors identify and characterize a novel chemical activator for selected members of the KCNQ/Kv7 channel family. The conclusions of the authors are overall well supported by a large amount of electrophysiology recordings, supplemented by a thorough mechanistic investigation using in silico studies, mutagenesis, and chemical synthesis. This contribution represents a significant discovery that will help move the field forward. The manuscript would be much improved after fixing several inconsistencies in data interpretation as well as few minor points described below:

Major points:

A “channel opener” effect logically exists if the stimulus alone is sufficient to significantly increase the channel macroscopic conductance above resting level. This seems to be the case with RAP against KCNQ3/5 (Fig 2I) and with CA against Q3* (Fig 3C, Fig 4) and KCNQ3/5 (Fig 7B). However, the weaker effect of RAP on Q1, Q2, Q3* and Q2/3 and the weaker effect of CA on Q5, Q2/5 and Q2/3/5 (Figure 7) does not justify, in my opinion, using the word “opener” to describe these effects, simply because conductance remains near zero at negative voltages. The authors should use different words to highlight the distinction between “opener” vs. weaker activatory effects.

This is interesting as we have previously been requested in other studies to refer to channel opening instead of activation. We have amended to “activate” or “negative-shifts the voltage dependence of” where appropriate, to clarify.

Page 7, 1st paragraph: The expression “under PiP2 replete conditions” is incorrect. There are only two conditions in Fig 5D: with Wortmannin (PiP2 depletion) or without Wortmannin (no change in PiP2). Using “replete” suggests PiP2 has been depleted first and then added back to the original level. This is not what happened according to the figure.

The definition of replete is “plentifully supplied or abounding in” – it doesn’t require prior depletion. However, we have amended it to “normal PIP2 levels” to avoid confusion.

Figure 8: The mutant phenotypes shown in Figure 8G-H should be compared with WT KCNQ3 data Figure 3A. Could the authors add reference G/Gmax curves (in both presence and absence of CA) to the G/Gmax plots in figure 8H? In addition, no discussion of phenotypes for other mutants (R239, M240, D241, G244) is given. I understand these mutations do not significantly alter the CA effects but this needs to be quantified or at least discussed.

Great point – we have added the reference curves to Figure 8. We have also added a discussion of the phenotypes of the mutations (page 9, paragraph 3).

Figure 9: Same as Figure 8: it would be nice to show reference G/Gmax curves for WT KCNQ3* to better visualize mutant phenotypes without going back and forth between figures.

We have also added the reference curves to Figure 9.

Page 9, 1st paragraph: The EC50 plot in Figure 8J (right panel) does not show a full sigmoid curve for the Q3*-R242A mutant and thus cannot be unambiguously interpreted. It seems that the red line is a fit to the data using some equation. If that is the case, the equation used for EC50 determination must be stated somewhere in the manuscript. If EC50 for CA is 0.1mM for this mutant (lower estimated boundary given by the authors), a symmetrical extrapolation of this fitted curve for saturation concentration of CA would give an Em shift of at least -60 mV (i.e. two times the -30mV shift observed at the EC50 value). This change would move Em from about -40 mV (control oocytes) to about -100 mV, which is below the K+ equilibrium potential (about -80 mV according to the reversal potential of K currents seen in most current traces). Because this theoretical value cannot be attained, the extrapolation of the fitted sigmoid curve cannot be used to estimate EC50. A second, more concerning problem with the estimated 100-fold decrease of EC50 is that it does not agree with the delta-V0.5 data from the left panel in the same figure: while the left panel shows a 2-fold decrease in efficacy (as correctly stated by the authors), there is little to no visible change in EC50. In my opinion, the data suggests that the mutation decreases maximal CA efficacy with no change in EC50.

We used a sigmoid fit but agree that it is not possible to unambiguously interpret the EM data nor attain an accurate midpoint value and have amended the text accordingly (page 9 paragraph 2).

Page 9, 2nd paragraph: “ten-fold more sensitive in term of potency” seems exaggerated. The middle panel in Figure 9C shows that the blue curve shifts by about 2/3 third of a decade in the log x-axis. A tenfold effect would have required a shift by a full decade. A 2/3 decade leftward shift corresponds to a 5-fold increase in EC50.

We have reduced our estimate to fivefold as suggested (page 10, paragraph 1).

Figure 10: In silico data suggests arginine mutations reduces binding but only weakly. There is no statistical error for the free energy calculations. Thus it is difficult to interpret this data. Regardless, the binding of CA remains strong in both WT and mutants. This data suggests that the mutants’ phenotype does not come from a decrease in binding free energy, as stated in the text. Can the authors comment on this?

We (or others) do not know how much binding energy reduction is needed to impair the ability of the compounds to activate the channel. We have added a discussion of this in the manuscript (page 14, final paragraph).

Minor points:

Fig 4H, x-axis label: “-120 mV” label in the voltage axis is missing (it is moved to panel G)

Moved back – thanks for spotting this.

Fig 5F, y axis title: “current decay” (not “deacy”)

Fixed, thank you.

Page 8, 2nd paragraph: Figure 8 A-C shows data from KCNQ5, not KCNQ3/5 as stated in the text.

This was from KCNQ3/5 – we have corrected the Figure 8 figure legend accordingly.

Page 11, 2nd paragraph: A negative sign should be put in front of binding free energies values

Corrected, thank you.

Page 14: I do not understand the clause: “..., and that both channel-bound and unbound PiP2 is depleted, without significant effects on CA opening ability”. The verb “is” refers to a singular subject but in this case, the subject “both channel bound- and unbound PiP2” should be plural. Do the authors mean to say something like: “..., and that regardless of the presence of PiP2, CA maintain its opening ability”? Please clarify this sentence.

You put it better than us – we have edited accordingly.

Page 15, 1st paragraph: The concluding sentence “...CA selectivity being functional rather than binding selectivity, i.e. in the effects of CS binding rather than the binding itself.” is difficult to parse. Grammatically, “rather than” in the first clause should be followed by an adjective (to oppose “functional”), not a noun (“binding selectivity”). The second explanatory clause does not help much. I get what the authors want to say in this sentence but only after deciphering its meaning. Rephrasing would make the idea clearer.

We have clarified by rephrasing to “consistent with carnosic acid selectivity being functional (i.e., binding opens some KCNQ isoforms but not others) rather than arising from its ability to bind per se”.

Page 15, 2nd paragraph: “holding the cell at -120mV”: this voltage is more negative than what a neurons can achieve in normal physiological conditions. This sentence does not align well within the rest of the paragraph dealing with in vivo/clinical applications.

We have clarified that that part of the discussion pertains to experimental aspects and reordered the paragraph to make that clearer.

Discussion: It would be nice if the authors could discuss hypothetical mechanisms by which channels with different subunit stoichiometry end up having different response to CA.

We have added a discussion on this (page 17 paragraph 1).

Reviewer #2 (Remarks to the Author):

The present work “ Ancient medicinal plant rosemary contains a highly efficacious and isoform selective KCNQ potassium channel opener” is an interesting study that aims to characterize the effect of natural

plant extracts on neuronal KCNQ channels with therapeutic potential. Using electrophysiology, mutagenesis, and in silico docking the authors showed that Carnosic acid (CA) –mainly– activates neuronal KCNQ3 and KCNQ3/5 channels over other KCNQ homo- and heteromeric complexes, somewhat underscoring CA-subunit selectivity. Overall, the experiments are nicely done and yield sound data that support the major claims of the paper. The manuscript is well structured and well written, and figures are well presented. The study proposes putative interaction sites of CA with KCNQ channels, which could be valuable to develop potential new neuroprotective drugs targeting KCNQ channelopathies. There are few suggestions that would further strengthen the report and a few other areas in need of further clarifications.

Major comments:

1) Page 7: “Here, depletion of PIP2 using pretreatment with wortmannin reduced KCNQ3* current magnitude ~fivefold, yet carnosic acid ‘CA’ (5 μ M) was still able to shift the KCNQ3* $V_{0.5act}$ (Figure 5A, B) and EM (Figure 5C) under these conditions, albeit the $\Delta V_{0.5act}$ was ~50% of that under PIP2 replete conditions (Figure 5D)”.

This sentence, and the data itself, is confusing and needs to be fixed. PIP2 is either depleted by Wortmannin and therefore KCNQ3 channel cannot open, which is not what the black-control trace in Fig 5A bottom panel shows (it shows only a reduction), or alternatively, Wortmannin only partially reduces PIP2 levels such that there is still a fraction of active KCNQ3 channels that can respond to CA. The conclusion that carnosic acid (5 μ M) can shift the KCNQ3* $V_{0.5act}$ ” even if PIP2 is ‘depleted’ needs to be revisited. The argument that Wortmannin reduces the current by 5-fold does not justify PIP2 ‘depletion’ whatsoever.

We have amended to “reduction”.

2) CA structure resembles more (although not quite) that of retigabine and ICA-069673 compounds than that of PIP2. How would CA and PIP2 compete for the same spot in the S4-S4-S5 loop? I would envision, much like for PIP2, that a drug acting in the proposed S4-S4-S5 linker interphase would behave like an amphipathic molecule. Is this the case of CA? This is not commented in the discussion. The authors did comment, however, in the discussion (page 14) that, “...carnosic acid does not require PIP2 and may even compete with PIP2 and replace it under some circumstances.” This is far too speculative...a suggestion simply lacking enough experimental support.

We have retained the discussion about the possible mechanisms, added a discussion about CA being amphipathic, but have removed the part about which model we favor, given the reviewer’s concern about sufficient supporting evidence even for a speculative discussion.

3) Along this line, the authors put forward the idea of CA interacting with the same PIP2-binding residues in the S4-S5 linker. Yet, PIP2 is believed to bind this pocket (but also the cytoplasmic ends of the VSD, the gate, helices A and B and their linkers, at least in homologous KCNQ2, 4 channels) in a state-dependent manner (only in the activated state, PMID: 24277843, PMID: 34650221, PMID: 29078287). Is CA only interacting with the S4-S5 linker and/or with all these other hotspots?... in a state dependent fashion? I think that the data provided in the current work is insufficient to formulate a precise picture of the

'binding' (maybe re-wording it to interacting sites, or effect?... See also below) site of CA onto KCNQ3 channels and more experiments need to be done to support such claim.

We don't think that the other PIP₂ binding sites are also CA binding sites (as pointed out, the chemistry between the two molecules is quite different, so not all interaction sites will be common to both). We do however believe there are likely other interacting residues close to the arginines that also contribute to the binding site and are embarking upon a comprehensive scanning mutagenesis study for a future manuscript. To maintain a balanced discussion we have softened "binding" to "interaction" in several instances.

4) Page 9, the authors state: "Overall, the data strongly suggest that KCNQ3 R242 and R243 are the most important components of the carnosic acid binding site, and likely directly coordinate carnosic acid binding". I recommend rephrasing this conclusion and reduce the tone of the claim. While the data in Figure 8 shows that Q3-R242A and R243A reduced the CA-mediated activation, especially when compared with the small effect shown by residues L198F, P211A, and W265L in Fig. 10, according to the tables of Figures 8-data, 100uM of carnosic acid still induced a GV shift of ~24 mV and a strong negative shift in Em in both Q3-R242A and Q3-R243A. While the data shown here suggests that R242 and R243 might participate in the CA activating effect, the evidence provided do not show that these residues are component of the binding site, much less that these residues directly coordinate CA binding. The authors convincingly show that CA effect on KCNQ3 is not through the retigabine and ICA/73 'binding' sites, which is already an important finding.

We have amended the sentence to "...that KCNQ3 R242 and R243 are important components of carnosic acid interaction and/or its functional effects." and have changed the header of that section to "*Carnosic acid interaction is facilitated by two arginines on the KCNQ3 S4-5 linker*"

5) Overall, the control traces for KCNQ3* (and KCNQ2/3, please compare Figs 2, g and 10) are not consistent throughout the figures, which raises concerns about the expression of the intended channel/subunit. For example, Fig. 10E, J, and R, looking at the traces denoted as KCNQ3* and Q3* in black (the nomenclature of KCNQ3* or Q3* ...should be also consistent) they all showed different kinetic of activation, particularly striking in the tail current. The same inconsistency is true for the G/Gmax V0.5 and Em values, which are all too variable, spanning a V-range from -50 to -40 mV in the V0.5 and -50 to -60 mV for the Em, in the example showed in carnosol and dimethylcarnosol, for instance. The same is true for instance in Figures 3C and 5B, G/Gmax, the V0.5 before Rosmarinic acid and before Wortmannin are -53 mV and -32 mV, respectively, which to me is beyond the permissive variable range. Furthermore, simple inspection of the GV curves (and V0.5) of KCNQ3* control (black trace) before carnosol in Fig.10F and methyl carnosate in Fig.10N, third and second graph from top, respectively, showed very different GV curves compared to similar control GV curves (e.g. for those before dimethylcarnosol, or any of the GVs shown for KCNQ3* in the entire paper). This is important because for instance, assuming that after correcting the (almost certain wrong) weird GV curve in Fig.10F the V0.5 is back to ~ 40mV (as shown throughout), then carnosol would then have induced a strong ~ 20mV hyperpolarizing shift of the GV, which would change the authors claims. These issues, and the concomitant conclusion derived from them need to be thoroughly revised throughout the ms. and an explanation needs to be provided.

The small changes in gating behavior can be explained by normal small fluctuations in oocyte properties over the several months it took to perform this study, are absolutely not unusual for this

type of study, and do not impact the results given that controls and corresponding test recordings are always made on the same oocyte within a few minutes of each other. We strongly disagree that these are consequential differences, or that they raise concerns over channel subunit expression. The control G/V curve in Figure 5B to which the reviewer is referring was recorded *after* wortmannin treatment (but before CA treatment), therefore the less negative V_{0.5} value is expected and completely appropriate. As for the figure 10F “weird” G/V curve...perhaps the review is referring to the KCNQ2/3 curve, which was only shown for the -80 to -20 mV range, so we have replaced it with one that spans the entire voltage range. Or perhaps the review is referring to the Q3* control for dimethyl carnosol, which is shown across a larger voltage range (down to -120 mV, which we do for some of the recordings) and hence looks slightly different due to the visual compression? We don’t agree with the contention that these G/V curves are unusual or consequentially different from others in the manuscript. In addition, given that independent groups were recorded over a period of several months (as is necessitated by a study with this many recordings) we don’t consider the spread of V_{0.5} and E_m values to be very big at all. Furthermore, the starting point within this small range does not change the magnitude of the V_{0.5} shifts with a particular compound, i.e., the compound-induced shifts remain consistent in our experience regardless of the starting point, in this G/V range (which we consider to be quite narrow given the volume of recordings and timeline in this project).

To check for consistency, we have recorded two new sets of data for KCNQ2/3 and for Q3*, each with versus without carnosol, dimethylcarnosol, carnosic diacetate, methyl carnosate, pisiferic acid, and/or carnosic γ lactone. There were no substantive differences between the new and old datasets and we have pooled the new data with the prior results, which we now show across the -80 to +40 mV range to allow for easier comparisons (Figure 10).

6) The authors state in the discussion: “Rosmarinic acid and carnosic acid are the rosemary phenolics most commonly associated with beneficial effects, largely attributed to their antioxidant and anti-inflammatory effects³⁴, but regulation of ion channels was not previously reported.” Yet, PMID: 29073181, for instance, reported that Rosemary extracts inhibited T-type Ca²⁺ (CaV_{3.2}) channel currents and left-shifted the steady-state inactivation of CaV_{3.2}. The authors should take advantage of this important report and discuss the synergistic effect of Rosemary extract on (activating) KCNQ and (inhibiting) Ca_{3.2} channels, particularly because Cav_{3.2}, much like KCNQ2/3, contributes to control neuronal excitability.

We missed the rosmarinic acid effects described in that paper, thank you for raising this omission. We now mention these in the Discussion (page 13, first paragraph).

7) In Fig. 7: what was the injected Q2/Q3/Q5 ratio? If it was 1:1:1, what is the composition of the ‘four’ missing subunit in the tetramer?

It was 1:1:1. Neither we nor others know the stoichiometry of the different isoforms in Q2/Q3/Q5 complexes. It would be an interesting future study to investigate this.

Minor comments

1) Fig.3 legend: “Mean unclamped oocyte membrane potential for KCNQ2-expressing oocytes as” This is wrong, right? Should not it be KCNQ3?

Corrected, thank you.

2) Furthermore, in Figures 3B, 6C, and 6F, Hesperidin: the black plot shows the peak-current/V but not the tail-current/V curve under control conditions, like for the other conditions...why?

We have replaced with the non-normalized tail current data for consistency.

3) Page 6: "...KCNQ2 insensitivity to carnosic acid is dominant in KCNQ2/3 heteromers". Figure 4J and S-Fig4-Table show that carnosic Acid cause around -10 mV shift of the G(V), so I would reword this here and maybe say instead "...KCNQ2 reduced sensitivity to carnosic acid" or something similar.

We have replaced with your suggested verbiage.

4) Along this line on Page 7, "...yet hesperidin did not open KCNQ2 (Figure 6D-F". It should be referred to Fig.6 E-G, right? Please, fix it.

Corrected.

5) Fig 5F, change deacy to decay.

Corrected.

Reviewer #3 (Remarks to the Author):

Thank you for the invitation to review "Ancient medicinal plant rosemary contains a highly efficacious and isoform selective KCNQ potassium channel opener" by Manville and colleagues. This is an interesting and comprehensive paper that describes the identification of carnosic acid in rosemary extract as a powerful activator of KCNQ3 subunits, with intermediate effects on KCNQ5, but minimal effects on KCNQ2. This leads to interesting selectivity towards different KCNQ heteromers, with relatively weak effects on Q2/Q3 heteromers, but strong potentiation of Q3/Q5 heteromers. Using mutagenesis, docking, and testing of chemical analogs of carnosic acid, the authors support a model of this compound binding in the vicinity of an arginine doublet at the intracellular end of the S4 segment. Overall, this study contains a significant amount of data and is quite convincing. I only have a few suggestions/questions for clarification.

Page 7 – 'dominant negative effects of KCNQ2' was a little confusing to me at first – perhaps describe as 'the insensitivity of KCNQ2 that dominated the response of KCNQ2/3 heteromers' or something like that.

We have edited as suggested.

Page 8... ' Thus carnosic acid likely underlies the KCNQ3/5 opening ability of heteromeric KCNQ3/5 channels.' I didn't understand the wording of this sentence, just a typo or something missed in the draft.

We have corrected our typo – the sentence now reads: "Thus, carnosic acid likely underlies the KCNQ3/5 opening ability of rosemary extract".

Page 8... 'Interestingly, the in silico docking predicted ionic bond formation between carnosic acid via its carboxylate group to the guanidinium group of R243 in KCNQ3, but not in KCNQ2'. Since these residues

are present in KCNQ2,3,4,5, perhaps the authors could clarify whether there are other contributors in this region that are candidates for the apparent selectivity? If there is nothing obvious, that's ok too as the functional outcome is so clear, but perhaps worth commenting.

We do not know; to clarify that this is unresolved, we have added the following: “We do not yet understand what about the KCNQ3 versus the KCNQ2 environment around R243 influences this predicted difference in interaction with carnosic acid.” And we discuss more in paragraph 1 on page 17.

Page 15... “our current data are consistent with carnosic acid selectivity being functional rather than binding selectivity, i.e., in the effects of carnosic acid binding rather than the binding itself”. Related to the previous comment/question – I was unclear what the authors were getting at here, so perhaps clarify. My impression earlier was that there was not predicted binding/interaction of carnosic acid to Q2. However, is this sentence implying binding with no effect?

There is a strong possibility that carnosic acid binds to Q2 but without effect. We have clarified this sentence, as follows: “our current data are consistent with carnosic acid selectivity being functional (i.e., binding opens some KCNQ isoforms but not others) rather than arising from its ability to bind per se.”

Overall this is an interesting study that will extend our understanding of KCNQ channel modulation, and highlights ways to modulate specific subtype combinations.

Thank you.

REVIEWERS' COMMENTS:

Reviewer #1 (Remarks to the Author):

The authors have addressed all my concerns appropriately.

Reviewer #2 (Remarks to the Author):

The authors have performed an impressive number of electrophysiological experiments that were combined with molecular modeling and medicinal chemistry to support their major claims of the paper. Overall, the authors have addressed most of my previous concerns.

Regarding point 5 from the previous round of review, the authors might still be "strongly" disagreeing with the reviewer and yet the variability among control experiments of channels of the same kind is still substantial. For instance, simply by inspecting Figure 10E, F control (black traces and black curves on carnosol and dimethylcarnosol, both the one I reviewed previously and the re-submitted one) one can see that:

1) the kinetic of the black traces are very different (please, compare the tails of the top and bottom black traces for a similar time scale). Although these traces are now amended in the present Fig. 10.
2) the $V_{0.5}$ from KCNQ2/3 (Fig. 10F second black curve) is somewhere between -45 and -50 mV, whereas the control for dimethylcarnosol (Fig. 10F fourth black curve) is close to -30 mV (e.g. the G/G_{max} at -40 mV in the top example is 80% of G_{max} , whereas in the bottom example it is around 30 % for the same -40 mV). These differences remain in current Fig.10.

The same is true in many other instances (e.g. Fig. 6 control Hesperidin, ursolic acid ($V_{0.5} \sim -30$ mV) vs control homoplantagin ($V_{0.5} \sim -50$ mV)...way more than 15 mV differences.

This is important considering the big effort and nice work the authors made in attributing specificity to some of these compounds for homo- vs heteromeric channels. Concerns remain about the composition of channel subunits attained after co-injecting 2/3, 2/5, 3/5...channels in a 1:1 ratio, which is not clear given the variability among control experiments.